# Personalized Federated Fine-tuning for Heterogeneous Data: An Automatic Rank Learning Approach via Two-Level LoRA

## Abstract

We study the task of personalized federated fine-tuning with heterogeneous data in the context of language models, where clients collaboratively fine-tune a language model (e.g., BERT, GPT) without sharing their local data, achieving personalization simultaneously. While recent efforts have applied parameter-efficient fine-tuning techniques like low-rank adaptation (LoRA) in federated settings, they typically use single or multiple independent low-rank adapters with predefined maximal and minimal ranks, which may not be optimal for diverse data sources over clients.

To address this issue, we propose PF2LoRA, a new personalized federated fine-tuning algorithm built on a novel *automatic rank learning approach via two-level LoRA*. Given the pretrained language model whose weight is frozen, our algorithm aims to learn two levels of adaptation simultaneously: the first level aims to learn a common adapter for all clients, while the second level fosters individual client personalization. A key advantage of PF2LoRA is its ability to adaptively determine a suitable rank based on an individual client's data, rather than relying on a predefined rank that is agnostic to data heterogeneity. We present a synthetic example that highlights how PF2LoRA automatically learns the ground-truth rank for each client, tailoring the adaptation to match the properties of their individual data. Notably, this approach introduces minimal additional memory overhead, as the second-level adaptation comprises a small number of parameters compared to the first level. Our experiments on natural language understanding and generation tasks demonstrate that PF2LoRA significantly outperforms existing federated fine-tuning methods.

## 1 Introduction

Federated learning (FL) (McMahan et al., 2017a; Kairouz et al., 2021) is a crucial paradigm for enabling collaborative training of machine learning models across distributed clients while preserving data privacy (McMahan et al., 2017b; Geyer et al., 2017). FL is particularly important in some scenarios that involve sensitive data, such as healthcare (Brisimi et al., 2018; Sheller et al., 2020), finance (Yang et al., 2019), and mobile devices (Bonawitz et al., 2019). However, in the context of foundation models like BERT (Devlin et al., 2018) and GPT (Radford et al., 2018), traditional FL algorithms face significant challenges due to the complexity of these models. It requires huge computing resources when directly fine-tuning model parameters on the heterogeneous data distributed across different clients.

To address the issue of fine-tuning foundation models, many parameter-efficient fine-tuning (PEFT) methods such as prompt tuning (Lester et al., 2021) and low-rank adaptation (LoRA) (Hu et al., 2021) have been explored, where LoRA freezes the original pre-trained parameters $W \in \mathbb{R}^{m \times n}$ of the foundation model while fine-tuning additional low rank matrices $B \in \mathbb{R}^{m \times r}$ and $A \in \mathbb{R}^{r \times n}$, $r \ll \min(m, n)$. This technique enables fine-tuning large models with a reduced number of trainable parameters, making them more suitable for resource-constrained devices. This paper specifically focuses on LoRA in the context of federated learning for heterogeneous data.

A natural method to perform low rank adaptation in federated learning is to adopt the same rank $r$ of matrices $A$ and $B$ across different clients. This method is referred to as homogeneous LoRA

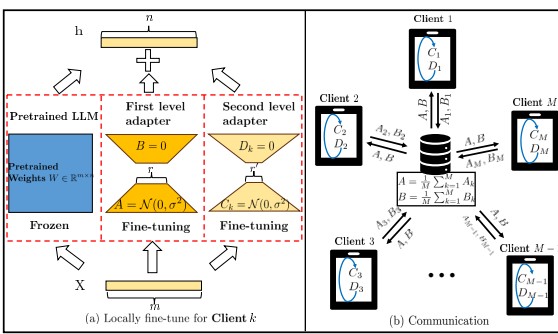

Figure 1: Overview of the two-level low-rank adaptation framework. The first level learns a common adapter $\{A, B\}$ for all clients, and the common adapter is synchronized by averaging across all the clients at every communication round. The second level aims to learn a client-specific and lightweight adapter $\{C_k, D_k\}$ for a specific client $k \in [1, M]$, while the lightweight adapters introduce negligible additional memory overhead.

(HOMLoRA), but it does not accommodate the personalized requirement of clients with heterogeneous data distributions. Recent work HETLoRA (Cho et al., 2024) highlights the importance of heterogeneous rank configurations to enable personalized federated learning, which proposed "matrix truncation", "local rank self-pruning", and "sparsity-weighted aggregation" to learn various ranks $r_k$ for the heterogeneous data from clients. However, this approach suffers from two main drawbacks: (1) The initial rank for any client is fixed and in the range of predefined minimal and maximal ranks, which is independent of client data. However, it is possible that the clients learning difficult tasks are assigned with smaller ranks and do not have the capacity to learn their corresponding tasks well. We empirically observe this issue and analysis theoretically the reason in Section 5. (2) There are many hyperparameters which need to be tuned, including the minimal and maximal values of rank, the pruning parameter, and the sparsity parameter. It remains unclear how to perform personalized federated fine-tuning such that the adapter is dependent on the data and the procedure has a small number of tuning parameters.

In this paper, we propose PF2LoRA, a novel personalized federated fine-tuning algorithm that explicitly incorporates heterogeneous ranks into the problem formulation. Our approach introduces a *two-level low-rank adaptation framework*. The first level learns a common adapter shared among all clients with trainable parameters $x = \{B \in \mathbb{R}^{m \times r}, A \in \mathbb{R}^{r \times n}\}$, while the second level enables client-specific personalization by learning lightweight, client-specific adapter $y_k$ for $k$-th client, defined as $y_k = \{D_k \in \mathbb{R}^{m \times \tilde{r}}, C_k \in \mathbb{R}^{\tilde{r} \times n}, 0 < \tilde{r} < r\}$ and $1 \leq k \leq M$ ($M$ represents the number of participating clients). We formulate the two-level low-rank adaptation framework as a bilevel optimization problem, aiming to learn a common adapter $x$ that minimizes the loss function given the fact the individual client adapters $\{y_k\}_{k=1}^{M}$ can achieve the optimal performance when conditioned on the shared adapter $x$. The two-level LoRA framework explicitly accommodates variations in adapter matrix ranks across clients, i.e. $r - \tilde{r} \leq r_k \leq r + \tilde{r}$. That allows the algorithm to automatically learn the ground-truth rank for each client based on their data heterogeneity.

Thus our algorithm essentially circumvents the rank pruning, matrix truncation, and zero-padding in HETLoRA for the alignment of adapters. Besides, the whole framework increases negligible additional memory overhead, as the second-level low rank adaptation comprises a small number of parameters compared to the first level. Our main contribution is listed as follows:

- We propose a novel bilevel formulation for personalized fine-tuning on heterogeneous data, and develop a two-level low rank adaptation framework to efficiently fine-tune foundation model in the scenario of federated learning. The main workflow of our framework is illustrated in Figure 1.

- We provide a synthetic example explaining why HETLoRA fails to learn the ground truth rank of clients, resulting in underfitting in a multivariate linear regression example. Then we conducted an experiment on personalized federated fine-tuning with two clients. The experimental results demonstrate that our algorithm can automatically learn the ground-truth of clients' rank to accomodate the data heterogeneity.

- Through extensive experiments on various natural language understanding and generation tasks, we demonstrate that PF2LoRA significantly outperforms existing federated fine-tuning baselines, providing a robust and efficient solution for personalized federated learning with foundation models. For example on GLUE benchmark, PF2LoRA achieves 25.6%, 2.33%, 15.34%, and 2.53% higher performance than state-of-the-art baseline HETLoRA on MNLI, SST-2, QQP, QNLI dataset, respectively. In addition, through extensive ablation studies, we show that our proposed two-level adaptation framework achieves the highest performance across various data heterogeneity levels and outperforms baseline methods even if they use more trainable parameters.

## 2 RELATED WORK

**Parameter-efficient Fine-Tuning.** There are various categories of parameter-efficient fine-tuning (PEFT) techniques, where only a subset of parameters of the pretrained foundation model or a small number of additional parameters are updated to adapt to the target task. The first line of work includes bias update or head-tuning (Lee et al., 2019; Zaken et al., 2021; Lawton et al., 2023; Wei et al., 2021) and weight masking (Zhao et al., 2020; Sung et al., 2021; Xu et al., 2021). The second line of work considers adapters (Houlsby et al., 2019; He et al., 2021a), prompt tuning (Lester et al., 2021; Li & Liang, 2021) and low rank matrix adaptation (Hu et al., 2021). Different from these works, our paper focuses on designing new federated learning algorithms based on low rank adaptation with heterogeneous data, where the local client data is not shared to other clients.

**Federated Learning with Fine-tuning.** The PEFT framework has been recently incorporated in the FL framework (Babakniya et al., 2023; Zhang et al., 2024; 2023b; Cho et al., 2024; Wang et al., 2023). However, most of them do not consider the data heterogeneity in the context of foundation models. To the best of our knowledge, HETLoRA (Cho et al., 2024) is the only work which allows data-independent heterogeneous ranks for each clients by a fixed rank initialization, zero-padding, truncation, self-pruning and sparsity regularization. In contrast, our work promotes data-dependent heterogeneous ranks of local clients by an explicit bilevel modeling and reduce the number of tuning hyperparameters.

## 3 PRELIMINARIES

In this section, we introduce a few parameter-efficient fine-tuning methods in the context of (federated) foundation model learning. It includes LoRA, HOMLoRA, HETLoRA. Due to limited space, we describe a variant of the personalized federated learning algorithm in the context of low rank adaptation method, namely Per-FedAvg-LoRA, in Appendix D.

**Low-rank adaptation (LoRA).** LoRA is a technique designed to efficiently fine-tune large pre-trained models by injecting trainable low-rank matrices into each layer of a foundation model (Hu et al., 2021). Formally, consider a pre-trained model where the original weight matrix is denoted as $W_0 \in \mathbb{R}^{m \times n}$. The model update $\Delta W$ during the fine-tuning can be approximated by multiplication of two low-rank matrices $B \in \mathbb{R}^{m \times r}$ and $A \in \mathbb{R}^{r \times n}$. The updated weight matrix $W$ is then given by:

$$W = W_0 + \Delta W = W_0 + BA. \tag{1}$$

This decomposition allows the model to learn adaptations for down-stream tasks while keeping the majority of the original weights frozen, thereby maintaining the pre-trained knowledge and significantly reducing memory and computational overhead.

**HOMLoRA.** When considering LoRA in the scenario of federated learning, a natural extension is refereed to as HOMLoRA, which adopts homogeneous rank $r$ across all the clients. Assume that $M$ clients participate in federated learning at every communication round. The objective function of each client $k$ is $f_k(\cdot)$, and the goal is to find a common adapter $x = \{A \in \mathbb{R}^{m \times r}, B \in \mathbb{R}^{r \times n}\}$ that performs well across all the clients. It aims to solve the optimization problem: $\min_x \frac{1}{M} \sum_{k=1}^{M} f_k(x)$. Specifically, each client locally updates their adapters for $I$ steps by Adam (or SGD) using their local data, and the server aggregates the adapters from each local clients $\{A_k^t, B_k^t\}_{k=1}^{M}$ ($k$ is the local client id) at iteration $t$ when $t$ is a multiple of $I$, where $I$ is the number of local updates per round: $A^t = \frac{1}{M} \sum_{k=1}^{M} A_k^t$, $B^t = \frac{1}{M} \sum_{k=1}^{M} B_k^t$. Then the server broadcasts the aggregated adapters back to each client. HoMLoRA can be regarded as a direct extension of FedAvg (McMahan et al., 2017a) in the context of LoRA (Hu et al., 2021).

**HETLoRA.** Recently, Cho et al. Cho et al. (2024) proposed a heterogeneous LoRA method, namely HETLoRA, which is able to learn heterogeneous low rank matrices for different clients. The main technical components contain four parts: (1) a fixed rank initialization: where the $r_k$ is fixed for $k$-th client and $r_{min} \leq r_k \leq r_{max}$; (2) distribution via truncation, wherein at each communication round, each client truncates the global adapter matrices to align dimensions $A_k^t = A_{:r_k,:}^t, B_k^t = B_{:,:r_k}^t$; (3) local training with self-pruning, which introduces the regularization term (with a penalty coefficient $\lambda$) to induce adapter sparsity (with a sparsity factor $\gamma$), and it dynamically reduces the $r_k$ by pruning unimportant columns in $B_k^t$ (or rows in $A_k^t$); (4) sparsity-weighted aggregation, wherein each communication round, to aggregate the adapter matrices with different rank $r_{min} \leq r_k \leq r_{max}$, the server reconstructs $\{A_k^t, B_k^t\}$ by zero-padding them to $r_{max}$.

Then HETLoRA updates the common adapter by aggregating the local adapters with an aggregation weight. In particular, the update rule is $A^{t+1} = \sum_{k=1}^M \|\Delta W_k^t\| A_k^t / \sum_{k=1}^M \|\Delta W_k^t\|$ and $B^{t+1} = \sum_{k=1}^M \|\Delta W_k^t\| B_k^t / \sum_{k=1}^M \|\Delta W_k^t\|$, $\Delta W_k^t = B_k^t A_k^t$.

However, the performance of HETLoRA heavily depends on (1) the fixed rank initialization, which is independent of data and may cause underfitting or overfitting issues, and (2) the proper setting for a set of hyperparameters, including $r_{\min}, r_{\max}, \gamma$, and $\lambda$. To address these issues, we propose a new two-level low-rank adaptation framework for personalized fine-tuning in the next subsection.

# 4 A New Two-level Adaptation for Personalized Federated Fine-Tuning

As we discussed in Section 3, HOMLoRA uses only one common adapter $x = \{B \in \mathbb{R}^{m \times r}, A \in \mathbb{R}^{r \times n}\}$ across all the clients, which is insufficient to learn from the heterogeneous data in federated learning. Therefore, we introduce a client-specific adapter for every client $k$ with $y_k = \{D_k \in \mathbb{R}^{m \times \tilde{r}}, C_k \in \mathbb{R}^{\tilde{r} \times n}, 0 < \tilde{r} < r, 1 \leq k \leq M\}$. We emphasize that the newly introduced adapter has a much smaller rank $\tilde{r}$ than that in the common adapter. Empirically, we usually set $\tilde{r} = \frac{r}{4}$ or $\frac{r}{2}$, which means the trainable parameters in the client-specific adapter are only $\frac{1}{4}$ or $\frac{1}{2}$ of those in the common adapter. Thus the new adapter is lightweight and incurs negligible additional memory overhead.

Different from (1), we incorporate both the common and client-specific adapters. In particular, the adapter for the $k$-th client can be parameterized as,

$$W_k = W_0 + BA + D_k C_k, \tag{2}$$

where $W_k$ is the adapter for $k$-th client, $A, B$ are common adapters for all clients, and $(C_k, D_k)$ are client-specific adapters for $k$-th client. Since the original weight $W_0$ is frozen, the trainable parameters in the model are $A, B, C_k, D_k$ for the client $k$. Different than the HETLoRA whose local client matrix rank is predefined and independent of data, our specific parameterization (2) explicitly encourages each adapter $W_k$ for the $k$-th client to vary over $k$: it can have different ranks in the range $(r - \tilde{r}, r + \tilde{r})$ and the specific rank is automatically determined by the training data.

We formalize our two-level adaptation framework for personalized federated fine-tuning as the following bilevel optimization problem:

$$\min_x \Phi(x) := \frac{1}{M} \sum_{k=1}^M f_k(x, y_k^*(x)), \text{(UL)} \quad \text{s.t.,} \quad y_k^*(x) \in \arg\min_{y_k} f_k(x, y_k), \text{(LL)} \tag{3}$$

where $f_k(x, y_k) := \mathbb{E}_{\xi \sim \mathcal{D}_k} F_k(x, y_k; \xi)$ is the loss function for the $k$-th client, $F_k$ the individual loss function for a sample $\xi$ from the $k$-th client, and $\mathcal{D}_k$ is the data on client $k$. The upper-level (UL) learns a common adapter $x$ for all the clients upon a set of the best client-specific adapters $\{y_k^*(x) \mid 1 \leq k \leq M\}$ for given $x$ defined by the lower-level problem. Given the common adapter, the lower-level (LL) aims to locally search the optimal client-specific adapter to fit its respective data, which in fact fosters individual client personalization.

**Algorithm Design.** Now we consider solving (3) efficiently in personalized federated learning. At the beginning of every communication round, i.e., $(t\%I = 0)$, each client $k$ receives the averaged common adapter $x_k^t$ from the server, and starts running its local steps. We run one step SGD for the lower-level problem to approximately find the minimizer of the lower-level problem (line 5 in Algorithm 1).

---

**Algorithm 1** TWO-LEVEL ADAPTATION FOR PERSONALIZED FINE-TUNING

---

1: **Input:** $\alpha, \eta, I, T, M, \mathcal{D}_k$
2: **for** $k \in \{1, \dots, M\}$ **in parallel do**
3:     **for** $t = 0, 1, \dots, T-1$ **do**
4:         Sample $\pi_k^t, \xi_k^t, \tilde{\xi}_k^t, \zeta_k^t$ independently from distribution $\mathcal{D}_k$
5:         $y_k^{t+1} = y_k^t - \alpha \nabla_y F_k(x_k^t, y_k^t; \pi_k^t)$
6:         $x_k^{t+1} = x_k^t - \eta \nabla_x F_k(x_k^t, y_k^{t+1}; \xi_k^t) + \alpha \eta \nabla_{xy} F_k(x_k^t, y_k^t; \zeta_k^t) \nabla_y F_k(x_k^t, y_k^{t+1}; \tilde{\xi}_k^t)$
7:         **if** $t\%I == 0$ **then**
8:             $x^{t+1} = \frac{1}{M} \sum_{k=1}^M x_k^{t+1}$
9:             $x_k^{t+1} = x^{t+1}$
10:         **end if**
11:     **end for**
12: **end for**

---

Define $\Phi_k(x) = f_k(x, y_k^*(x))$, then the gradient of the function $\Phi_k(x_k^t)$ in terms of $x_k^t$, namely hypergradient (Ghadimi & Wang, 2018), can be calculated by chain rule approximately as follows:

$$\nabla \Phi_k(x_k^t) \approx \nabla \widehat{\Phi}_k(x_k^t) = \nabla_x f_k(x_k^t, y_k^{t+1}) - \alpha \nabla_{xy} f_k(x_k^t, y_k^t) \nabla_y f_k(x_k^t, y_k^{t+1}), \quad (4)$$

where $\approx$ is due to the fact that $y_k^{t+1}$ is only an approximation to the optimal solution $y_k^*(x_k^t)$. Therefore, we use the stochastic version of $\nabla \widehat{\Phi}_k(x_k^t)$ to update the common adapter $x$ on client $k$, as described in line 6 of Algorithm 1.

In fact, Adam or AdamW can also be used to update the upper-level variable based on the stochastic gradient information to replace the SGD update as in line 6. Empirically, we adopt AdamW as the upper-level optimizer (line 6) and SGD as the lower-level optimizer (line 5) to fine-tune a language model. One can refer to Algorithm 1 for more details, where line 5 is used to update the client-specific adapter, line 6 is used to update the common adapter, and line 8 corresponds to the synchronization of the common adapter.

## 5 AUTOMATIC RANK LEARNING

To clarify this mechanism of "automatic rank learning of PF2LoRA", as well as the failure reason of HETLoRA, we first construct a multivariate linear regression example and provide a theoretical analysis to demonstrate why our method is able to learn the ground-truth rank, accommodating the heterogeneity of clients' data, whereas HETLoRA fails. Then we conduct a synthetical experiment to compare two algorithms in federated learning with two clients. The experimental results confirm that our algorithm is able to learn and converge to the optimal solution. In contrast, HETLoRA underestimates the initial rank of some clients due to random rank initialization strategy, resulting in underfitting and suboptimal performance in such clients.

Consider a multivariate linear regression in federated learning,

$$\min_W \sum_{k=1}^2 \|X_k W - Y_k\|_F^2$$

where $(X_k, Y_k)$ is the client-$k$'s data, $W$ is a low-rank matrix and can be decomposed into low-rank matrices, $W = AB$. The details of synthetic experiments are described as follows,

**Ground truth of trainable parameters.** Given two clients, suppose that we have two optimal solutions with low-rank structure,

$$W_1^* = A_1^* B_1^*, \; s.t., W_1^* \in \mathbb{R}^{10 \times 10}, A_1^* \in \mathbb{R}^{10 \times 3}, B_1^* \in \mathbb{R}^{3 \times 10},$$

$$W_2^* = A_2^* B_2^*, \; s.t., W_2^* \in \mathbb{R}^{10 \times 10}, A_2^* \in \mathbb{R}^{10 \times 4}, B_2^* \in \mathbb{R}^{4 \times 10},$$

with $rank(W_1^*) = 3$, $rank(W_2^*) = 4$. We initialize the random matrices $A_1^*, B_1^*, A_2^*, B_2^* \sim \mathcal{N}(0, 1)$[1].

---

[1] Note that we use $X \sim \mathcal{N}(0, 1)$ to denote each entry of the matrix $X$ follows a standard Gaussian distribution.

**Training and testing data.** We construct the synthetic data $(X, Y)$ for two clients respectively by randomly generating 1000 samples, i,e., $X_1 \in \mathcal{R}^{1000 \times 10}, s.t., X_1 \sim \mathcal{N}(0, 1)$, $X_2 \in \mathcal{R}^{1000 \times 10}, s.t., X_2 \sim \mathcal{N}(0, 1)$, and their element targets,

$$y_1 = x_1 W_1^* + \epsilon_1, \ \epsilon_1 \sim \mathcal{N}(0, 0.1), \quad y_2 = x_2 W_2^* + \epsilon_2, \ \epsilon_2 \sim \mathcal{N}(0, 0.2).$$

The first 700 samples serve as the training set $\mathcal{D}_k^{tr}, k = 1, 2$ and the remaining serves as the testing set $\mathcal{D}_k^{te}, k = 1, 2$.

**Training process.** HETLoRA: Following its rank initialization strategy $r_{min} \leq rank_1 \leq rank_2... \leq rank_k... \leq r_{max}$, we assume that $r_{min} = 1, r_{max} = 12$ and initialize $\hat{W}_k = \hat{A}_k \hat{B}_k$ by,

$$\hat{A}_1 \in \mathbb{R}^{10 \times 2}, \hat{B}_1 \in \mathbf{0}^{2 \times 10}, \ s.t. \ \hat{A}_1 \sim \mathcal{N}(0, 1), \quad \hat{A}_2 \in \mathbb{R}^{10 \times 10}, \hat{B}_2 \in \mathbf{0}^{10 \times 10}, \ s.t. \ \hat{A}_2 \sim \mathcal{N}(0, 1)$$

so we have $rank(\hat{A}_1) = 2$ and $rank(\hat{A}_2) = 10$. We can easily get that the total number of trainable parameters for two clients is 240.

PF2LoRA: For a fair comparison, we initialize the trainable parameters $\hat{W}_k = \hat{A}_k \hat{B}_k + \hat{C}_k \hat{D}_k$, and make sure the total number of trainable parameters to be the same as that in HETLoRA. For client $k = 1, 2$, we have $r = 4, \tilde{r} = 2$ and,

$$\hat{A}_k \in \mathbb{R}^{10 \times 4}, \hat{B}_k \in \mathbb{R}^{4 \times 10}, \hat{C}_k \in \mathbb{R}^{10 \times 2}, \hat{D}_k \in \mathbb{R}^{2 \times 10},$$

$$s.t. \ \hat{A}_k, \hat{C}_k, \hat{C}_k, \hat{D}_k \sim \mathcal{N}(0, 1).$$

and $A_k B_k$ is orthogonal to the matrix $C_k D_k$, such that their column space or row space are independent mutually. The total number of training steps are fixed as 2000, and the communication interval is 10. The details of hyperparameter settings, inlcuding learning rate, pruning parameter of HETLoRA etc., are summarized in Appendix C.

**Evaluation.** We evaluate the model at each communication round on the test set $\mathcal{D}_k^{te}$, and track the Frobenius distance $\|\hat{W}_k - W_k^*\|_F^2$ along with the rank evolution of each client. For PF2LoRA, we compute the singular values $\{\lambda_i | i = 1, ..., 10\}$ of $\hat{W}_k$ via SVD and define the rank as the smallest $j$ satisfying $\min_{1 \leq j \leq 10} \sum_{i=1}^{j} \lambda_i \geq 0.9 \sum_{i=1}^{10} \lambda_i$, with $\{\lambda_i\}$ in descending order. Figure 2 shows comparisons on training/testing loss, Frobenius distance, and rank evolution.

In particular, PF2LoRA accurately recovers the ground-truth ranks of 3 and 4 for the two clients, validating its ability to automatically adapt ranks within $[r - \tilde{r}, r + \tilde{r}]$. It achieves near-zero loss and small distance to $W_k^*$. In contrast, HETLoRA fails to learn the correct rank for client 1 due to an underestimated initialization, further exacerbated by rank pruning to $r_{\min} = 1$, leading to higher loss and deviation. Client 2 benefits from a better initialization and successfully recovers its rank. See Appendix B for theoretical support.

# 6 EXPERIMENTS

We evaluate PF2LoRA and baseline methods on two major NLP tasks: natural language understanding (NLU) and natural language generation (NLG), where NLU experiments include the text classification on GLUE benchmark (Wang et al., 2018) and question answering task on SQuAD v1 (Rajpurkar et al., 2016) and v2 (Rajpurkar et al., 2018). NLG experiments are performed on E2E NLG Challenge dataset (Novikova et al., 2017) and WebNLG dataset (Gardent et al., 2017). Then we execute the ablation studies to explore (1) the performance comparison when other baselines have more trainable parameters than ours in Appendix I; (2) the impact of data heterogeneity in Appendix I.1; and (3) the role of bilevel optimization in our framework in Appendix I.2. Training stability is further analyzed in Appendix J. Baseline methods include Centralized LoRA, Homogeneous LoRA (HOMLoRA), Personalized Federated Average LoRA (Per-FedAvg-LoRA), and Heterogeneous LoRA (HETLoRA). The parameter sensitivity analysis is deferred to Appendix K, and the computing and communication costs are presented in Appendix L.

## 6.1 NATURAL LANGUAGE UNDERSTANDING

### 6.1.1 ROBERTA ON TEXT CLASSIFICATION

**Model**. We adopt RoBERTa (Liu et al., 2019) as the backbone for personalized federated fine-tuning on GLUE tasks, using both the base (125M) and large (355M) versions with LoRA. While baselines

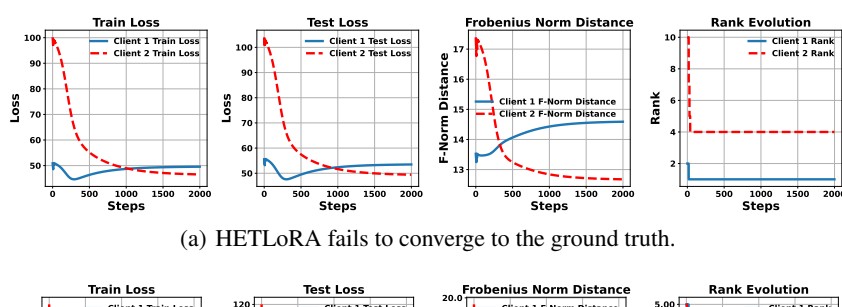

(a) HETLoRA fails to converge to the ground truth.

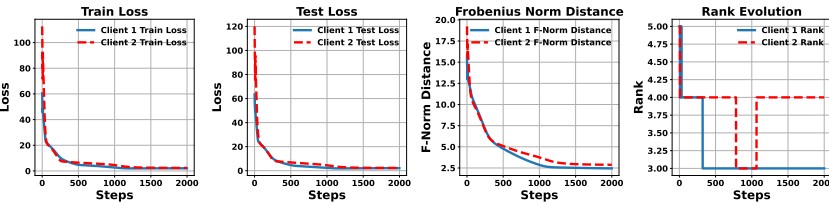

(b) PF2LoRA can converge to the ground truth.

Figure 2: Comparison of two algorithms. Left to right: the training loss on two clients, the testing loss on two clients, Frobenius norm distance $\|W_k - W_k^*\|_F$, $k = 1, 2$, and the rank evolution of two clients.

Table 1: RoBERTa-base results on GLUE.

| Method | CoLA | MNLI | SST-2 | QQP | QNLI |
|---|---|---|---|---|---|
| Centralized LoRA | 56.85 | 83.48 | 93.58 | 86.97 | 89.70 |
| HOMLoRA | 50.75 | 70.56 | 92.47 | 79.61 | 85.45 |
| Per-FedAvg-LoRA | 51.11 | 74.73 | 90.56 | 81.26 | 78.59 |
| HETLoRA | 53.76 | 73.33 | 93.67 | 81.49 | 91.86 |
| PF2LoRA | **54.19** | **92.14** | **95.85** | **93.99** | **94.18** |

Table 2: DeBERTa-v3 results on SQuAD.

| Method | SQuAD v1.0 (EM/F1) | SQuAD v2.0 (EM/F1) |
|---|---|---|
| Centralized LoRA[2] | 68.72/83.36 | 44.56/53.31 |
| HOMLoRA | 68.57/82.99 | 42.53/52.70 |
| Per-FedAvg-LoRA | 68.80/83.08 | 43.15/53.16 |
| HETLoRA | 68.64/83.28 | 44.53/54.69 |
| PF2LoRA | **71.61/85.11** | **44.95/54.71** |

inject only common adapters into attention layers, PF2LoRA additionally includes lightweight, client-specific adapters. The adapter ranks are configured as follows:

1. Centralized LoRA, HOMLoRA, and Per-FedAvg-LoRA: fixed rank $r_k = 8$ for all clients.

2. HETLoRA: client-specific ranks $r_k$ initialized as $r_{min} + \frac{(r_{max} - r_{min})(k-1)}{M}$.

3. PF2LoRA: common adapter rank $r_k = 8$, client-specific rank $\tilde{r}_k = 2$.

Rank initialization and trainable parameter counts are detailed in Table 6 (Appendix F.1). PF2LoRA introduces a slight increase in parameter count. Since HETLoRA assigns different ranks to different clients, we report the average number of trainable parameters across clients. Results with higher-parameter baselines are discussed in Section I.

**Dataset**. Following the non-i.i.d. partitioning protocol in (Karimireddy et al., 2020), we split datasets into heterogeneous client subsets based on a similarity parameter $s \in [0, 1]$. Each client's data contains $(100 \times s)\%$ i.i.d. samples and $100 \times (1-s)\%$ label-sorted samples from the full dataset. We evaluate on five GLUE classification tasks: CoLA, MNLI, SST-2, QQP, and QNLI. Dataset statistics are provided in Table 7 (Appendix F.1).

**Experiment Details**. We run federated fine-tuning across 8 clients (NVIDIA A100 GPUs), with all clients participating and client 0 responsible for parameter aggregation and distribution each round. Centralized LoRA, HOMLoRA, and HETLoRA use AdamW to update the common adapter. Per-FedAvg-LoRA applies SGD for the one-step local update and AdamW for the global adapter. PF2LoRA uses SGD for client-specific adapters and AdamW for the common adapter. Learning rates are tuned individually, with optimal values listed in Table 5 (Appendix F.1). For fair comparison, we set the batch size to $\mathcal{B} = 16$ and the communication interval to $I = 10$ across all federated baselines. The total number of communication rounds $R$ varies by dataset: CoLA (50), MNLI (300), SST-2 (100), QQP (300), and QNLI (100), and is kept consistent across methods.

We execute evaluate model on each client's test data and report averaged results. Metrics include "Matthews's correlation" for CoLA and "Accuracy" for MNLI,SST-2, QQP, QNLI. Results are presented in Table 1 (RoBERTa base) and Table 13 (RoBERTa large) in Appendix G, where the heterogeneity level $s = 0.3$ is set for CoLA and $s = 0.9$ for MNLI, SST-2, QQP, and QNLI. PF2LoRA outperforms all baselines significantly on MNLI, SST-2, QQP, QNLI, and achieves comparable performance to Centralized LoRA on CoLA while surpassing other federated baselines.

### 6.1.2 DeBERTa on Question Answering

**Model**. We use DeBERTa (He et al., 2021b), an enhanced transformer encoder with 86M parameters, for question-answering tasks on SQuAD v1 and v2. DeBERTa improves text understanding compared to BERT and RoBERTa, making it suitable for complex tasks like question answering and sentiment analysis.

**Dataset**. SQuAD v1/v2 are reading comprehension datasets containing over 100k and 150k question–answer pairs, respectively, extracted from Wikipedia. In v1, all questions have answers in the passage, whereas v2 includes unanswerable questions, increasing task difficulty. The training set covers 442 unique topics; test sets include 48 (v1) and 35 (v2) topics. We sample $80\%$ of the original training set for training and use the remaining $20\%$ for testing. Heterogeneous client data is constructed by topic using the method in Section 6.1.1 with heterogeneity parameter $s = 0.5$. Evaluation is based on Exact Match (EM) and F1 scores.

**Experiment Details**. Due to the complexity of QA tasks, we perform federated fine-tuning on 4 clients (NVIDIA A100 GPUs) using heterogeneity level $s = 0.5$, with 200 communication rounds ($R = 200$) and a communication interval of $I = 10$. Optimizers follow Section 6.1.1, and the batch size is fixed at $\mathcal{B} = 16$ for fair comparison. Optimal learning rates and rank settings are detailed in Tables 8 and 9 (Appendix F.2). As shown in Table 2, PF2LoRA achieves the highest EM and F1 scores across all federated baselines-outperforming the best baseline by $4.08\%$ (EM) and $2.20\%$ (F1) on SQuAD v1.

### 6.2 Natural Language Generation

For NLG tasks, we follow LoRA (Hu et al., 2021) to use GPT-2 medium model for federated fine-tuning on WebNLG and E2E NLG Challenge dataset.

**Model**. We use GPT-2 Medium (345M) and GPT-2 XL (1.5B) (Radford et al., 2019) for NLG experiments.

**Dataset**. WebNLG spans 10 domains (sports, cities, universities, hotels, etc.). We create 8 client partitions, keeping each client's train/test domains consistent. E2E targets restaurants; we split 34 training and 18 test restaurants into 8 client groups by name so each client's test set only includes restaurants seen in its training.

Table 3: GPT-2 generation results on WebNLG dataset.

| Method | BLEU ↑ | MET ↑ | TER ↓ | ROUGE-L ↑ |
|---|---|---|---|---|
| Centralized LoRA | 0.6031 | 0.7807 | 0.5900 | 0.4169 |
| HOMLoRA | 0.5141 | 0.7271 | **0.5697** | 0.4736 |
| Per-FedAvg-LoRA | 0.5152 | 0.7219 | 0.5746 | 0.4740 |
| HETLoRA | 0.5196 | 0.7219 | 0.5746 | 0.4740 |
| PF2LoRA | **0.5261** | **0.7301** | 0.5733 | **0.4769** |

Table 4: The comparison results with more trainable parameters in baselines. We report "Matthew's correlation" for CoLA and "Accuracy" for MNLI, SST-2, QQP and QNLI. Higher value means "better performance".

| Method | # Param | CoLA | MNLI | SST-2 | QQP | QNLI |
|---|---|---|---|---|---|---|
| HOMLoRA | 0.44M | 52.01 | 73.82 | 92.63 | 80.11 | 86.27 |
| Per-FedAvg-LoRA | 0.44M | 52.35 | 78.62 | 89.65 | 81.12 | 81.41 |
| HETLoRA | 0.43M | 53.43 | 79.32 | 94.83 | 81.71 | 92.12 |
| PF2LoRA | **0.37M** | **54.19** | **92.14** | **95.85** | **93.99** | **94.18** |

---

[2]Results differ from (Zhang et al., 2023a) due to harder test data with 442 topics vs. 48.

**Experiment Details**. Following LoRA (Hu et al., 2021), we (i) fine-tune adapters, (ii) generate with beam search, (iii) decode, and (iv) evaluate. We use 8 clients (each on an NVIDIA A100), where each client fine-tunes on its own domain (WebNLG) or restaurant subset (E2E) and evaluates on its own test set. Metrics: BLEU, NIST, METEOR, TER, ROUGE-L, CIDEr.

We run $R=200$ rounds on WebNLG and $R=300$ on E2E with interval $I=10$, batch size $\mathcal{B}=4$, and beam width $bw=10$. Optimizers follow Sec. 6.1.1; tuned step sizes are in Tables 10, 11 (Appendix F.3). Rank settings and parameter counts are in Table 12. Results for GPT-2 Medium are in Tables 3, 14 (Appendix. H); GPT-2 XL results are in Table 15. PF2LoRA is best on nearly all metrics, e.g., $+1.25\%$ BLEU on WebNLG and $+3.85\%$ BLEU on E2E than HETLoRA.

## 7 THEORETICAL JUSTIFICATION

In this section, we provide the theoretical justification for the Algorithm 1 in an simplified scenario: we consider the single machine case ($M = 1$) and assume we have access to the deterministic gradient oracle. In this case the algorithm reduces the following formulation:

$$\min_x \Phi(x) \coloneqq f(x, y^*(x)) \text{ (UL)}, \quad \text{s.t., } y^*(x) \in \arg\min_y f(x, y) \text{ (LL)}, \tag{5}$$

The update of Algorithm 1 in the single machine case with deterministic gradient reduces to the following update rule:

$$
\begin{aligned}
y^{t+1} &= y^t - \alpha \nabla_y F(x^t, y^t) \\
x^{t+1} &= x^t - \eta[\nabla_x F(x^t, y^{t+1}) + \alpha \nabla_{xy} F(x^t, y^t) \nabla_y F(x^t, y^{t+1})].
\end{aligned}
\tag{6}
$$

We will establish the convergence of the update rule (6) under the following assumptions.

**Assumption 7.1.** (i) $f$ are bounded below, $\Phi(x_0) - \min_x \Phi(x) \le \Delta$; (ii) $f$ is $\mu$-strongly convex in terms of $y$ for given $x$ ; (iii) $f$ is continuously differentiable and $L_{f,1}$-smooth jointly in $(x,y)$; (iv) $f$ is twicely differentiable and $\nabla^2 f$ is $L_{f,2}$-Lipschitz jointly in $(x, y)$.

**Remark**: These assumptions are standard in the bilevel optimization literature (Kwon et al., 2023; Ji et al., 2021).

**Theorem 7.2** (Convergence Guarantees). *Suppose Assumption 7.1 holds. Define the smoothness parameter* $L_\Phi = L_{f,1} + \frac{L_{f,1}^2}{\mu}$, *and choose* $\alpha = \frac{1}{4L_{f,1}}, \eta =$ $\min\left(\frac{\mu^2}{5L_{f,1}^3\sqrt{(\frac{4L_{f,1}}{\mu} - \frac{\mu}{4L_{f,1}})}}, \frac{1}{8L_\Phi}, \sqrt{\frac{1}{16N}}, \sqrt[3]{\frac{1}{81NL_\Phi}}\right),$ *and* $N = \frac{25L_{f,1}^4(\frac{4L_{f,1}}{\mu}+1)}{16\mu^2}$. *Then, we have* $\frac{1}{T}\sum_{t=0}^{T-1} \|\nabla\Phi(x^t)\|^2 \le O(1/T)$, *where $T$ is the total number of iterations.*

**Remark**: Theorem 7.2 provides a convergence guarantee with $O(1/T)$ convergence rate for the squared gradient norm. It means that it requires $O(1/\epsilon^2)$ gradient or Hessian-vector product evaluations for finding an $\epsilon$-stationary point (i.e., finding a $x$ such that $\|\nabla\Phi(x)\| \le \epsilon$). This complexity matches the convergence rate of gradient descent for smooth nonconvex function. In addition, compared with existing double-loop bilevel optimization algorithms such as Ji et al. (2021), our update rule (6) is an single-loop bilevel algorithm and hence is easy to implement in practice.

## 8 CONCLUSION

In this paper, we presented PF2LoRA, a novel personalized federated fine-tuning algorithm for heterogeneous data based on a two-level LoRA framework, where the first level aims to learns a common adapter for all the clients and the second level fosters individual client personalization. Our approach achieves automatic rank learning and addresses the limitations of existing methods, such as data-independent rank initialization and excessive hyperparameter tuning. Through comprehensive experiments on NLU and NLG tasks, PF2LoRA demonstrated significant performance improvements over state-of-the-art baselines, with negligible additional memory overhead. While our method is designed for language models, its applicability to other modalities like vision or multimodal learning has yet to be explored, which we leave as future work.

REPRODUCIBILITY STATEMENT

We provide Assumption 7.1 and Theorem 7.2 in main text and proofs of Theorem 7.2 in Appendix A. An anonymized code with training/evaluation scripts, configurations, seeds, and environment files is included in the supplementary materials. All base models are publicly available: RoBERTa, DeBERTa, and GPT2 (used under MIT license). Datasets GLUE, SQuAD (under CC BY-SA 4.0 license), WebNLG (under CC BY-NC 4.0 license) and E2E NLG (under CC BY-SA 4.0 license) are accessible on HuggingFace under the licenses stated on their corresponding Hugging Face dataset cards. We include download scripts, preprocessing/splits. These materials sufficiently support the reproduction of our results.

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

## A    PROOF OF THEOREM 7.2

### A.1    BASIC LEMMAS

The hypergradient estimation is defined as $\nabla\widehat{\Phi}(x; y^{t+1}) = \nabla_x f(x, y^{t+1}) - \alpha\nabla_{xy}f(x, y^t)\nabla_y f(x, y^{t+1})$.

**Lemma A.1** (gradient descent for strongly convex and smooth functions). *when $\alpha \leq \frac{1}{L_{f,1}}$, for lower level each step we have*

$$\|y^{t+1} - y^*(x^t)\| \leq (1 - \alpha\mu)^{\frac{1}{2}}\|y^t - y^*(x^t)\|. \tag{7}$$

*Proof.* Note that

$$\|y^{t+1} - y^*(x^t)\|^2 = \|y^t - \alpha\nabla_y f(x^t, y^t) - y^*(x^t)\|^2 \tag{8}$$

$$= \|y^t - y^*(x^t)\|^2 - 2\alpha\langle\nabla_y f(x^t, y^t), y^t - y^*(x^t)\rangle + \alpha^2\|\nabla_y f(x^t, y^t)\|^2$$

$$\overset{(i)}{\leq} (1 - \alpha\mu)\|y^t - y^*(x^t)\|^2 - 2\alpha(f(x, y^t) - \inf_y f(x^t, y)) + \alpha^2\|\nabla f_y(x^t, y^t)\|^2$$

$$\overset{(ii)}{\leq} (1 - \alpha\mu)\|y^t - y^*(x^t)\|^2 - 2\alpha(f(x^t, y^t) - \inf_y f(x^t, y)) + 2\alpha^2 L_{f,1}(f(x^t, y^t)) - \inf_y f(x^t, y))$$

$$= (1 - \alpha\mu)\|y^t - y^*(x^t)\|^2 - 2\alpha(1 - \alpha L_{f,1})(f(x^t, y^t) - \inf_y f(x^t, y))$$

$$\overset{(iii)}{\leq} (1 - \alpha\mu)\|y^t - y^*(x^t)\|^2 \tag{9}$$

where $(i)$ is because of the $\mu$-strongly convexity, $(ii)$ is because of $L_{g,1}$-smooth of the function, $(iii)$ is because of $2\alpha(1 - \alpha L_{f,1})(f(x^t, y^t) - \inf_y f(x^t, y)) \geq 0$.  □

**Lemma A.2** (true hypergradient). *The hypergradient $\nabla\Phi(x)$ equals to $\nabla_x f(x, y^*(x))$.*

*Proof.* By the implicit function theorem (Ghadimi & Wang, 2018), we have

$$\nabla\Phi(x) = \nabla_x f(x, y^*(x)) - \nabla_{xy}f(x, y^*(x))[\nabla_{yy}f(x, y^*(x))]^{-1}\nabla_y f(x, y^*(x)) \overset{(i)}{=} \nabla_x f(x, y^*(x))$$

where $(i)$ holds due to $\nabla_y f(x, y^*(x)) = 0$.  □

**Lemma A.3** (Lipschitz property (Ghadimi & Wang, 2018)). *$y^*(x)$ is $\frac{L_{f,1}}{\mu}$-Lipschitz continuous.*

**Lemma A.4** (Lipschitz hypergradient). *$\Phi(x)$ is $L_\Phi$-smooth and $L_\Phi = L_{f,1} + \frac{L_{f,1}^2}{\mu}$.*

*Proof.* By definition of hypergradient in Lemma A.2 and Assumption 7.1, we have

$$\|\nabla\Phi(x_1) - \nabla\Phi(x_2)\| = \|\nabla_x f(x_1, y^*(x_1)) - \nabla_x f(x_2, y^*(x_2))\|$$

$$\leq L_{f,1}\|x_1 - x_2\| + L_{f,1}\|y^*(x_1) - y^*(x_2)\|$$

$$\overset{(i)}{\leq} L_{f,1}\|x_1 - x_2\| + \frac{L_{f,1}^2}{\mu}\|x_1 - x_2\| = L_\Phi\|x_1 - x_2\|, \tag{10}$$

where $(i)$ comes from Lemma A.3.  □

### A.2    PROOF

**Lemma A.5** (Hypergradient bias). *Hypergradient estimation $\nabla\widehat{\Phi}(x; y^{t+1})$ satisfy:*

$$\|\nabla\widehat{\Phi}(x^t; y^{t+1}) - \nabla\Phi(x^t)\| \leq L_{f,1}(\alpha L_{f,1} + 1)(1 - \alpha\mu)^{\frac{1}{2}}\|y^t - y^*(x^t)\|$$

*Proof.* Note that

$$\nabla\widehat{\Phi}(x^t; y^{t+1}) - \nabla\Phi(x^t)$$
$$= \nabla_x f(x^t, y^{t+1}) - \alpha\nabla_{xy}f(x^t, y^t)\nabla_y f(x^t, y^{t+1}) - \nabla_x f(x^t, y^*(x^t))$$
$$\overset{(i)}{=} \nabla_x f(x^t, y^{t+1}) - \nabla_x f(x^t, y^*(x^t)) - \alpha\nabla_{xy}f(x^t, y^t)(\nabla_y f(x^t, y^{t+1}) - \nabla_y f(x^t, y^*(x^t)))$$
(11)

where $(i)$ holds due to $\nabla_y f(x^t, y^*(x^t)) = 0$. Then we obtain that

$$\|\nabla\widehat{\Phi}(x^t; y^{t+1}) - \nabla\Phi(x^t)\|$$

$$\overset{(i)}{\leq} (L_{f,1} + \alpha L_{f,1}^2)\|y^{t+1} - y^*(x^t)\|$$

$$\overset{(ii)}{\leq} (L_{f,1} + \alpha L_{f,1}^2)(1 - \alpha\mu)^{\frac{1}{2}}\|y^t - y^*(x^t)\|$$
(12)

$$= A\|y^t - y^*(x^t)\|$$
(13)

where $A = (L_{f,1} + \alpha L_{f,1}^2)(1 - \alpha\mu)^{\frac{1}{2}}$, $(i)$ holds because $\nabla_x f$ and $\nabla_y f$ are $L_{f,1}$ Lipschitz with $x, y$, and $(ii)$ holds due to Lemma A.1. □

**Lemma A.6** (Hypergradient descent). *Define $A = (L_{f,1} + \alpha L_{f,1}^2)(1 - \alpha\mu)^{\frac{1}{2}}$, we have*

$$\frac{1}{T}\sum_{t=0}^{T-1}(\frac{1}{2} - \eta L_\Phi)\|\nabla\Phi(x^t)\|^2 \leq \frac{\Phi(x_0) - \inf\Phi(x)}{\eta T} + \frac{1}{T}(\frac{1}{2} + \eta L_\Phi)A^2\sum_{k=0}^{T-1}\|y^t - y^*(x^t)\|^2 \quad (14)$$

*Proof.* The proof is very similar to the proof of Theorem 1 in Ji et al. (2021). The $L_\Phi$-smoothness of $\Phi(x)$ implies that

$$\Phi(x^{t+1}) - \Phi(x^t) \leq \langle\nabla\Phi(x^t), x^{t+1} - x^t\rangle + \frac{L_\Phi}{2}\|x^{t+1} - x^t\|^2 \quad (15)$$

Define $h^t = \nabla\widehat{\Phi}(x^t; y^{t+1}) = \nabla_x f(x^t, y^{t+1}) - \alpha\nabla_{xy}f(x^t, y^t)\nabla_y f(x^t, y^{t+1})$. We have

$$\Phi(x^{t+1}) \leq \Phi(x^t) - \eta\langle\nabla\Phi(x^t), h^t\rangle + \frac{L_\Phi\eta^2}{2}\|h^t\|$$

$$\leq \Phi(x^t) - \eta(\frac{1}{2} - \frac{\eta L_\Phi}{2})\|h^t\|^2 + \frac{\eta^2 L_\Phi}{2}\|h^t - \nabla\Phi(x^t)\|^2$$

$$\leq \Phi(x^t) - (\frac{\eta}{2} - \eta^2 L_\Phi)\|\nabla\Phi(x^t)\|^2 + (\frac{\eta}{2} + \eta^2 L_\Phi)\|h^t - \nabla\Phi(x^t)\|^2 \quad (16)$$

Do telescoping and use Lemma A.5 we get

$$\frac{1}{T}\sum_{t=0}^{T-1}(\frac{1}{2} - \eta L_\Phi)\|\nabla\Phi(x^t)\|^2 \overset{\text{Lemma } A.5}{\leq} \frac{\Phi(x_0) - \inf\Phi(x)}{\eta T} + \frac{1}{T}(\frac{1}{2} + \eta L_\Phi)A^2\sum_{k=0}^{T-1}\|y^t - y^*(x^t)\|^2$$
(17)

□

**Lemma A.7** (Lower Level Convergence). $\|y^{t+1} - y^*(x^{t+1})\|^2 \leq C\|y^t - y^*(x^t)\|^2 + D\|\nabla\Phi(x^t)\|^2$, *where* $C = 1 - \alpha^2\mu^2 + 2(1 + \frac{1}{\alpha\mu})\frac{L_{f,1}^2}{\mu^2}\eta^2 A^2$, $D = 2(1 + \frac{1}{\alpha\mu})\eta^2\frac{L_{f,1}^2}{\mu^2}$.

*Proof.* Note that

$$\|y^{t+1} - y^*(x^{t+1})\|^2$$

$$\overset{(i)}{\leq} (1 + \alpha\mu)\|y^{t+1} - y^*(x^t)\|^2 + (1 + \frac{1}{\alpha\mu})\|y^*(x^{t+1}) - y^*(x^t)\|^2$$

$$\overset{(ii)}{\leq} (1 + \alpha\mu)(1 - \alpha\mu)\|y^t - y^*(x^t)\|^2 + (1 + \frac{1}{\alpha\mu})\frac{L_{f,1}^2}{\mu^2}\|x^{t+1} - x^t\|^2$$

$$\leq (1 + \alpha\mu)(1 - \alpha\mu)\|y^t - y^*(x^t)\|^2 + (1 + \frac{1}{\alpha\mu})\frac{2L_{f,1}^2}{\mu^2}\eta^2(\|h^t - \nabla\Phi(x^t)\|^2 + \|\nabla\Phi(x^t)\|^2)$$

$$= C\|y^t - y^*(x^t)\|^2 + D\|\nabla\Phi(x^t)\|^2, \quad (18)$$

where $(i)$ uses the Young's inequality, $(ii)$ is due to Lemma A.1 and the Lipschitzness of the mapping $y^*(x)$, $C = 1 - \alpha^2\mu^2 + 2(1 + \frac{1}{\alpha\mu})L_y^2\eta^2A^2$; $D = 2(1 + \frac{1}{\alpha\mu})\eta^2\frac{L_{f,1}^2}{\mu^2}$. $\qquad\square$

*Proof of Theorem 7.2.* Substituting Lemma A.7 to Lemma A.6 yields

$$\frac{1}{T}\sum_{t=0}^{T-1}\left[\frac{1}{2} - \eta L_\Phi - (\frac{1}{2} + \eta L_\Phi)A^2D\right]\|\nabla\Phi(x^t)\|^2$$

$$\leq \frac{\Phi(x^0) - \inf\Phi(x)}{\eta T} + \frac{1}{T}(\frac{1}{2} + \eta L_\Phi)A^2\frac{\|y^0 - y^*(x^0)\|^2}{1 - C}, \qquad (19)$$

where $A = (L_{f,1} + \alpha L_{f,1}^2)(1 - \alpha\mu)^{\frac{1}{2}}$, $C = 1 - \alpha^2\mu^2 + 2(1 + \frac{1}{\alpha\mu})\frac{L_{f,1}^2}{\mu^2}\eta^2A^2$; $D = 2(1 + \frac{1}{\alpha\mu})\eta^2\frac{L_{f,1}^2}{\mu^2}$.

We want to carefully choose the parameter $\alpha, \eta$ s.t. $C < 1$, $\alpha \leq \frac{1}{L_{f,1}}$ and $\frac{1}{2} - \eta L_\Phi - (\frac{1}{2} + \eta L_\Phi)A^2D > 0$. For example, we can choose $\alpha = \frac{1}{4L_{f,1}}, \eta = \min\left(\frac{\mu^2}{5L_{f,1}^3\sqrt{(\frac{4L_{f,1}}{\mu} - \frac{\mu}{4L_{f,1}})}}, \frac{1}{8L_\Phi}, \sqrt{\frac{1}{16N}}, \sqrt[3]{\frac{1}{81NL_\Phi}}\right)$, and $N = \frac{25L_{f,1}^4(\frac{4L_{f,1}}{\mu}+1)}{16\mu^2}$.

$\qquad\square$

# B    THEORETICAL ANALYSIS: AN EXAMPLE ON MULTIVARIATE LINEAR REGRESSION

In this section, we provide the theoretical analysis to demonstrate why our method is able to learn the ground truth rank, whereas HETLoRA fails in a multivariate linear regression example

If our algorithm can find a better low rank approximation than HETLoRA, then our method surely performs better than HETLoRA. So theoretically, we want to find the exact analytic solution of the best low rank approximation. Recall multivariate linear regression problem, the goal is to minimize the reconstruction error:

$$\min_{W\in\mathbb{R}^{m\times n}}\|Y - XW\|_F^2$$

where $(X, Y)$ is the data and label. We know the solution which can minimize the reconstruction error is,

$$W = (X^TX)^{-1}X^TY$$

However, $rank(W)$ is possibly very large, leading to computationally inefficient. So we want to find the optimal low-rank matrix approximation of $W$ (i.e. minimize the reconstruction error with small rank of $W$), then we add a rank restriction on $W$,

$$Y = XW + \epsilon, \quad \text{s.t.,} \quad rank(W) \leq r.$$

In statistics, this is a Reduced Rank Regression (RRR) problem, which has been well-explored,

$$\min_{W\in\mathbb{R}^{m\times n}}\|Y - XW\|_F^2, \quad \text{s.t.,} \quad rank(W) \leq r,$$

which is equivalent to

$$\min_{W\in\mathbb{R}^{m\times n}}tr[(Y - XW)(Y - XW)^T], \quad rank(W) \leq r$$

where $tr(.)$ is the matrix trace.
Given the upper bound of $rank(W) = r$, we directly do rank factorization on $W$, i.e., LoRA:

$$\min_{A\in\mathbb{R}^{m\times r}, B\in\mathbb{R}^{r\times n}}tr[(Y - XAB)(Y - XAB)^T],$$

Specifically in HETLora setting, given the rank initialization of the $k-$client: $r_k^{init}$, the objective function is:

$$\min_{A\in\mathbb{R}^{m\times r_k^{init}}, B\in\mathbb{R}^{r_k^{init}\times n}}tr[(Y_k - X_kAB)(Y_k - X_kAB)^T].$$

In our setting, we initialize the rank of the common adapter to $r$, and the local adapter to $\tilde{r}$, the objective function is,

$$\min_{A\in\mathbb{R}^{m\times r}, B\in\mathbb{R}^{r\times n}, C_k\in\mathbb{R}^{m\times\tilde{r}}, D_k\in\mathbb{R}^{\tilde{r}\times n}} tr[(Y_k - X_k(AB+C_kD_k))(Y_k - X_k(AB+C_kD_k)^T].$$

In the synthetic experiment, we make global $AB$ to be in the orthogonal row and vector space of $C_kD_k$, then we directly get

$$r(W_k) = r(AB + C_kD_k) = r(AB) + r(C_kD_k) = r + \tilde{r}$$

then our problem is equivalent to reduced-rank regression problem.

**Lemma B.1.** *(Reinsel & Velu, 1998) Theorem 2.2[RRR solution] Suppose the $(m+n)$-dimensional random vector $(Y_k, X_k)$ has mean vector $0$ and covariance matrix with:*

$$\Sigma_{yx} = \Sigma_{xy} = Cov(Y_k, X_k) \quad and \quad \Sigma_{xx} = Cov(X_k) \quad nonsingular.$$

*Then, for any positive-definite matrix $\Sigma$, an $m \times r$ matrix $A$ and $r \times n$ matrix $B$, for $r \leq \min(m,n)$, which minimize*

$$tr\{\mathbb{E}[\Sigma^{1/2}(Y_k - X_kAB)(Y_k - X_kAB)^\top\Sigma^{1/2}]\}$$

*are given by:*

$$A^{(r)} = \Sigma^{-1/2}[V_1,\ldots,V_r] = \Sigma^{-1/2}V, \quad B^{(r)} = V^\top\Sigma^{1/2}\Sigma_{yx}\Sigma_{xx}^{-1}$$

*where $V = [V_1,\ldots,V_r]$ and $V_j$ is the (normalized) eigenvector that corresponds to the $j$-th largest eigenvalue $\lambda_j^2$ of the matrix:*

$$\Sigma^{1/2}\Sigma_{yx}\Sigma_{xx}^{-1}\Sigma_{xy}\Sigma^{1/2}, \quad j = 1, 2, \ldots, r.$$

From solution formula we directly get minimum truncated error

$$\min_{A,B:rank(AB)\leq r} \|W - AB\|_F^2 = \sqrt{\sum_{i=r+1}^{n}\lambda_i} \quad \forall W, rank(W) \geq r$$

### B.0.1 LOW-RANK APPROXIMATION

Specifically in HETLoRA setting, given the rank initialization of the $k-$client: $r_k^{init}$, the objective function is:

$$\min_{A\in\mathbb{R}^{m\times r_k^{init}}, B\in\mathbb{R}^{r_k^{init}\times n}} tr[(Y_k - X_kAB)(Y_k - X_kAB)^T].$$

In our setting, we initialize the rank of the common adapter to $r$, and the local adapter to $\tilde{r}$, the objective function is,

$$\min_{A\in\mathbb{R}^{m\times r}, B\in\mathbb{R}^{r\times n}, C_k\in\mathbb{R}^{m\times\tilde{r}}, D_k\in\mathbb{R}^{\tilde{r}\times n}} tr[(Y_k - X_k(AB+C_kD_k))(Y_k - X_k(AB+C_kD_k)^T]. \quad (20)$$

note $C_kD_k$, is a local adapter. we mark

$$W_k = P_kQ_k = AB + C_kD_k$$

note that $rank(P_kQ_k) \in [r-\tilde{r}, r+\tilde{r}]$. Generally we cannot say the problem (20) and

$$\min_{P_k\in\mathbb{R}^{m\times r+\tilde{r}}} Q_1\in\mathbb{R}^{r+\tilde{r}\times n} tr[(Y_k - X_kP_kQ_k)(Y_k - X_kP_kQ_k)]^T,$$

are equivalent since the former one is subset of the latter problem. However, under some certain dataset setting, the two problems are equivalence. We defer the equivalence proof to Lemma B.0.2.

Suppose we have two clients, the optimal solution in HETLoRA is

$$\text{Client 1} \quad A_1^{r_1^{init}} = \Sigma^{-1/2}[V_1,\ldots,V_{r_{init}}] = \Sigma^{-1/2}V, \quad B_1^{r_1^{init}} = V^\top\Sigma^{1/2}\Sigma_{yx}\Sigma_{xx}^{-1}$$

$$\text{Client 2} \quad A_2^{r_2^{init}} = \Sigma^{-1/2}[V_1, \ldots, V_{r_{init}}] = \Sigma^{-1/2}V, \quad B_2^{r_2^{init}} = V^\top \Sigma^{1/2} \Sigma_{yx} \Sigma_{xx}^{-1}$$

In our setting, the optimal solution is

$$\text{Client 1} \quad P_1^{r+\tilde{r_1}} = \Sigma^{-1/2}[V_1, \ldots, V_{r+\tilde{r_1}}] = \Sigma^{-1/2}V, \quad Q_1^{r+\tilde{r_1}} = V^\top \Sigma^{1/2} \Sigma_{yx} \Sigma_{xx}^{-1}$$

$$\text{Client 2} \quad P_2^{r+\tilde{r_2}} = \Sigma^{-1/2}[V_1, \ldots, V_{r+\tilde{r_2}}] = \Sigma^{-1/2}V, \quad Q_1^{r+\tilde{r_1}} = V^\top \Sigma^{1/2} \Sigma_{yx} \Sigma_{xx}^{-1}$$

Suppose for Client 1 data, $\Sigma^{1/2}\Sigma_{yx}\Sigma_{xx}^{-1}\Sigma_{xy}\Sigma^{1/2}$ has eigenvector $\lambda_1 = \lambda_2 = \lambda_3 = 1; \lambda_4 = \cdots = \lambda_n = 0$, obviously the low rank approximation is $r_1^* = 3$. For Client 2 data, $\Sigma^{1/2}\Sigma_{yx}\Sigma_{xx}^{-1}\Sigma_{xy}\Sigma^{1/2}$ has eigenvector $\lambda_1 = \cdots = \lambda_4 = 1; \lambda_5 = \cdots = \lambda_n = 0$, the low rank approximation is $r_2^* = 4$.

In our synthetic experiments 5, HETLoRA underestimates the rank for client 1, i.e., $r_1^{init} = 2 < r_1^* = 3$ due to the random rank initialization, and the learned rank $r_1 = 1$ by self-pruning; Client 2 initializes a reasonable $r_2^{init} = 10$, and the learned rank $r_2 = 5 = r_2^*$. Thus client 1 fails to learn the optimal low rank approximation because

$$\min_{A,B:rank(AB)\leq r_1^{init}} \|W - AB\|_F^2 = \sqrt{\sum_{i=r+1}^n \lambda_i} = 1.$$

Our PF2LoRA initializes $r = 4$ for the common adapter $(AB)$, and $\tilde{r} = 2$ $(C_k D_k)$ for the local adapter, which means $r - \tilde{r} = 2 \leq rank(AB + C_k D_k) \leq r + \tilde{r} = 6$, and learned rank for client 1 is $r_1 = 3$. The learned rank for client 2 is $r_2 = 4$. Both succeeded to learn the optimal low rank approximation.

$$\min_{A,B,C_k,D_k:r-\tilde{r}\leq rank(W_k)\leq r+\tilde{r}} \|W - W_k\|_F^2 = \sqrt{\sum_{i=r+\tilde{r}}^n \lambda_i} = 0.$$

### B.0.2 PROBLEM EQUIVALENCE

Next we prove two problems to be equivalent:

$$\min_{A\in\mathbb{R}^{m\times r} \quad B\in\mathbb{R}^{r\times n} \quad C_k\in\mathbb{R}^{m\times\tilde{r}} \quad D_k\in\mathbb{R}^{\tilde{r}\times n}} tr[(Y_k - X_k(AB + C_k D_k))(Y_k - X_k(AB + C_k D_k)^T]$$

and

$$\min_{W_k\in\mathbb{R}^{m\times n}} tr[(Y_k - X_k W_k)(Y_k - X_k W_k)^T], \quad r - \tilde{r} \leq rank(W_k) \leq r + \tilde{r}$$

**Lemma B.2.** *The rank of the sum of $AB$ and $CD$ satisfies:*

$$r(AB + CD) = r(AB) + r(CD)$$

*if and only if*

$$dim(\mathcal{C}_1 \cap \mathcal{C}_2) = dim(\mathcal{R}_1 \cap \mathcal{R}_2) = 0.$$

*where $\mathcal{C}_1$ and $\mathcal{C}_2$ be the column spaces of $AB$ and $CD$, and $\mathcal{R}_1$, $\mathcal{R}_2$ are their row spaces.*

*Proof.* To simplify the notation in proof, we mark $c = dim(\mathcal{C}_1 \cap \mathcal{C}_2)$, $d = dim(\mathcal{R}_1 \cap \mathcal{R}_2)$; $E = AB$, $F = CD$. First, the condition $c = d = 0$ is necessary, as two strings of inequalities show:

$$r(E + F) \leq r[(E, F)] = r(E) + r(F) - c \leq r(E) + r(F),$$

$$r(E + F) \leq r[(E; F)] = r(E) + r(F) - d \leq r(E) + r(F).$$

To show $c = d = 0$ is sufficient, we use full-rank decompositions of $E$ and $F$:

$$E = C_1 R_1, \quad r(A) = r(C_1) = r(R_1) = a,$$

where $E$ is $m \times n$, $C_1$ is $m \times a$, and $R_1$ is $a \times n$.

$$F = C_2 R_2, \quad r(F) = r(C_2) = r(R_2) = b,$$

where $F$ is $m \times n$, $C_2$ is $m \times b$, and $R_2$ is $b \times n$.

Such representations exist since $R_1$ can be any matrix whose rows form a basis of the row space of $A$. Then $A = C_1 R_1$ for some $C_1$, and:

$$r(E) = r(C_1) = \min\big(\text{rank}(C_1), \text{rank}(R_1)\big) \leq a = r(E).$$

We now write:

$$E + F = C_1 R_1 + C_2 R_2 = (C_1, C_2)\begin{pmatrix} R_1 \\ R_2 \end{pmatrix} = CR,$$

Then $c = 0$ implies that all the $a + b$ columns of $C$ are linearly independent, and so $C$ has a left inverse $L$ such that $LC = I$. Thus, when $c = 0$,

$$r(E + F) = r(CR) \geq r(LCR) = r(R) = r(E) + r(F) - d.$$

If in addition $d = 0$, the entire string collapses, and:

$$r(E + F) = r(E) + r(F).$$

$\square$

In the following synthetic experiment setting we make global $AB$ in orthogonal row and vector space of $C_1 D_1, C_2 D_2$, according to above lemma we directly get

$$r(W_1) = r(AB + C_1 D_1) = r(AB) + r(C_1 D_1) = r + \tilde{r}$$

and

$$r(W_2) = r(AB + C_2 D_2) = r(AB) + r(C_2 D_2) = r + \tilde{r}$$

So under our synthetic experiment setting, our problem is equivalent to reduced-rank regression problem, which provides a theoretical guarantee.

## C  EXPERIMENTAL SETTINGS IN THE SYNTHETIC EXAMPLE

We conduct a synthetic experiment of multivariate linear regression in federated learning to show why HETLoRA fails to learn the ground truth rank, but PF2LoRA does. The following describes the details of experiments and the hyperparameter settings for both algorithms,

1. HETLoRA: Following its rank initialization strategy $r_{min} \leq rank_1 \leq rank_2 ... \leq rank_k ... \leq r_{max}$, we assume that $r_{min} = 1, r_{max} = 12$ and initialize $\hat{W}_k = \hat{A}_k \hat{B}_k$ by,

$$\hat{A}_1 \in \mathbb{R}^{10 \times 2}, \hat{B}_1 \in \mathbf{0}^{2 \times 10}, \ s.t. \ \hat{A}_1 \sim \mathcal{N}(0,1),$$

$$\hat{A}_2 \in \mathbb{R}^{10 \times 10}, \hat{B}_2 \in \mathbf{0}^{10 \times 10}, \ s.t. \ \hat{A}_2 \sim \mathcal{N}(0,1)$$

so we have $rank(\hat{A}_1) = 2$ and $rank(\hat{A}_2) = 10$. We can easily get that the total number of trainable parameters for two clients is 240. Other hyperparameters are set as follows. We search the regularization factor $\gamma$ in the range $[0.05, 0.5]$ with the search grid $0.05$ and set it to the optimal value $0.1$. The pruning parameter $\gamma = 0.3$, which is responsible for imposing the regularization to the last $30\%$ columns to sparse them. We tune the learning rate within the range $\{0.001, 0.002, 0.003, 0.004, 0.005\}$ and set it to the optimal value $0.002$. The total training steps are 2000, and the communication is performed every 10 steps, which means we train the parameters for 10 steps locally, and then execute the parameter aggregation and distribution.

2. PF2LoRA: For a fair comparison, we initialize the trainable parameters $\hat{W}_k = \hat{A}_k \hat{B}_k + \hat{C}_k \hat{D}_k$, and make sure the total number of trainable parameters to be the same as that in HETLoRA. For client $k = 1, 2$, we have $r = 4, \tilde{r} = 2$ and,

$$\hat{A}_k \in \mathbb{R}^{10 \times 4}, \hat{B}_k \in \mathbb{R}^{4 \times 10}, \hat{C}_k \in \mathbb{R}^{10 \times 2}, \hat{D}_k \in \mathbb{R}^{2 \times 10},$$

$$s.t. \ \hat{A}_k \sim \mathcal{N}(0,1), \hat{C}_k \sim \mathcal{N}(0,1), \hat{C}_k \sim \mathcal{N}(0,1), \hat{D}_k \sim \mathcal{N}(0,1).$$

and $A_k B_k$ is orthogonal to the matrix $C_k D_k$, such that their column space or row space are independent mutually. The total number of training steps are fixed as 2000, and the communication interval is 10. We search the best upper-level and lower-level learning rates within the range $[0.001, 0.01]$ with the search grid of $0.001$, and set the best upper-level learning rate to $0.005$ and the lower-level learning rate to $0.002$. In each communication round, we aggregate the common adapter parameters $A_k, B_k$ and then distribute them, and the local adapter parameters $C_k, D_k$ are not involved in communication.

## D    DETAILS OF PER-FEDAVG-LORA

**Per-FedAvg-LoRA**. Per-FedAvg-LoRA is built upon a well-known personalized federated learning approach called Per-FedAvg (Fallah et al., 2020), with the trainable model parameters being low rank matrices such as in LoRA. Per-FedAvg is a typical personalized federated learning algorithm, which incorporates Model-Agnostic Meta-Learning (MAML) (Finn et al., 2017) to FedAvg algorithm (McMahan et al., 2017a) to enable models quickly adapting to heterogeneous data. When it is applied to low rank adaptation, we can get a new variant, namely Per-FedAvg-LoRA. The goal of Per-FedAvg-LoRA is to find a common adapter $x$ which can perform well after it is updated by one-step gradient descent on each client. In particular, Per-FedAvg-LoRA is trying to solve the following formulation using the FedAvg algorithm:

$$\min_x \frac{1}{M} \sum_{k=1}^{M} f_k(x - \alpha \nabla f_k(x)), \tag{21}$$

where $\alpha > 0$ is the step size. Note that Per-FedAvg-LoRA uses adapter matrices with homogeneous rank across all the clients.

## E    PYTORCH-STYLE PSEUDOCODE FOR PF2LORA

In this section, we show the PyTorch-style pseudocode for PF2LoRA. Our two-level low rank adapter framework can be derived by slightly modifying the LoRA module and integrating it into federating learning. When creating low rank adapters, we need to initialize two types of adapters, i.e., common adapters and the client adapters. The initial rank dimension for the common adapter is typically set to $r$, while for the client adapter, it is set to $\frac{r}{2}$. In addition, we require two different optimizers to update the common and client adapters. The common adapter is updated using AdamW, and the client adapter is updated using SGD. It's important to note that hypergradient calculation is necessary when updating the common adapter. Besides, our framework can be easily plugged into multiple language models, such as RoBERTa, DeBERTa and GPT-2, and others.

## F    EXPERIMENT SETUP

### F.1    ROBERTA ON TEXT CLASSIFICATION

We use grid search to find the best learning rate for each algorithm in the range of $\{1.0 \times 10^{-4}, 5.0 \times 10^{-4}, 1.0 \times 10^{-3}, 2.0 \times 10^{-3}, 5.0 \times 10^{-3}\}$. For algorithm Per-FedAvg-LoRA, we search the optimal learning rate for one-step update and the common adapter update, respectively. For PF2LoRA, we also search for the best learning rate for the client-specific adapter update and the common adapter update. The selected learning rates for each algorithm are listed in Table 5, where we use slash to separate two learning rates for Per-FedAvg-LoRA and PF2LoRA, with the former learning rate being for the common adapter. For HETLoRA, we fix the sparsity parameter $\gamma = 0.99$ across all the datasets and set the penalty factor $\lambda = 1.0 \times 10^{-3}$ for CoLA dataset, and $\lambda = 5.0 \times 10^{-3}$ for MNLI, SST-2, QQP, and QNLI. The rank initialization and the number of trainable parameters are summarized in Table 6. The details of the text classification datasets are summarized in Table 7.

Table 5: Learning rate setting for RoBERETa model on GLUE benchmark. We use slash to separate two learning rates for Per-FedAvg-LoRA and PF2LoRA. For Per-FedAvg-LoRA, the former one is the learning rate for the common adapter, the latter one is the learning rate for one-step SGD. For PF2LoRA, the former one is the learning rate for the common adapter, the latter one is the learning rate for the client-specific adapter.

| Method | CoLA | MNLI | SST-2 | QQP | QNLI |
|---|---|---|---|---|---|
| Centralized LoRA | $1.0 \times 10^{-3}$ | $1.0 \times 10^{-3}$ | $1.0 \times 10^{-3}$ | $1.0 \times 10^{-3}$ | $1.0 \times 10^{-3}$ |
| HOMLoRA | $1.0 \times 10^{-3}$ | $1.0 \times 10^{-3}$ | $2.0 \times 10^{-3}$ | $1.0 \times 10^{-3}$ | $1.0 \times 10^{-3}$ |
| Per-FedAvg-LoRA | $2.0 \times 10^{-3}/1.0 \times 10^{-2}$ | $1.0 \times 10^{-3}/1.0 \times 10^{-3}$ | $1.0 \times 10^{-3}/1.0 \times 10^{-3}$ | $1.0 \times 10^{-3}/1.0 \times 10^{-3}$ | $2.0 \times 10^{-3}/1.0 \times 10^{-3}$ |
| HETLoRA | $5.0 \times 10^{-3}$ | $2.0 \times 10^{-3}$ | $2.0 \times 10^{-3}$ | $2.0 \times 10^{-3}$ | $2.0 \times 10^{-3}$ |
| PF2LoRA | $2.0 \times 10^{-3}/1.0 \times 10^{-4}$ | $1.0 \times 10^{-3}/1.0 \times 10^{-3}$ | $1.0 \times 10^{-3}/1.0 \times 10^{-3}$ | $1.0 \times 10^{-3}/1.0 \times 10^{-3}$ | $1.0 \times 10^{-3}/1.0 \times 10^{-3}$ |

**Algorithm 2** PyTorch-style Pseudocode for PF2LoRA

```
1  # model_name: the name of pretrained model
2  # lr_in, lr_out: the learning rate for client and common adapter
3  # T: the total number of communication rounds, I: communication
       interval
4  # r: low rank parameter
5  # train_dataloader
6
7  import torch.distributed as dist
8  dist.init_process_group()
9  target_modules = ["query", "value"]
10 pretrained_model = LLM_Model.from_pretrained(model_name)
11 model = get_peft_model(pretrained_model, target_modules, r)
12 optimizer_outer = AdamW(model.common_adpter.parameters(), lr_in)
13 optimizer_inner = SGD(model.client_adpter.parameters(), lr_out)
14
15 step = 0
16 for epoch_idx in range(total_epochs)
17     for data_batch in train_dataloader:
18         inner_batch, outer_batch = data_batch
19         update_client_adapter(model, inner_batch, optimizer_inner)
20         update_common_adapter(model, outer_batch, optimizer_outer)
21         if step % I == 0:
22             dist.reduce(model.common_adapter.parameters(), dst=0,
23          op=self.dist.ReduceOp.SUM)
24             average(model.common_adpter.parameters())
25             dist.broadcast(model.common_adapter.parameters(), src=0)
26         step += 1
27 #
28 def get_peft_model(pretrained_model, target_modules, r)
29     for module_name, _ in pretrained_model.named_modules():
30         if module_name in target_modules:
31             target_module= pretrained_model.get_submodule(module_name
       )
32             create_and_replace(target_module, r)
33
34 def create_and_replace(target_module, r)
35     if isinstance(target_module, Linear):
36         target_module.initialize_common_adapter(r)
37         target_module.initialize_client_adapter(r/2)
38         target_module.set_trainable_params()
```

Table 6: Trainable parameters of RoBERTa-base/large.

| Method | # Trainable Parameters (base/large) |
|---|---|
| HOMLoRA ($r_k = 8$) | 0.30M/0.79M |
| Per-FedAvg-LoRA ($r_k = 8$) | 0.30M/0.79M |
| HETLoRAS ($r_{max} = 12, r_{min} = 8$) | 0.35M/0.94M |
| PF2LoRA ($r_k = 8, \tilde{r}_k = 2$) | 0.37M/0.99M |

## F.2 DEBERTA ON QUESTION ANSWERING

We search for the optimal learning rate from the range of $\{1.0 \times 10^{-4}, 5.0 \times 10^{-4}, 1.0 \times 10^{-3}, 2.0 \times 10^{-3}, 5.0 \times 10^{-3}, 1.0 \times 10^{-2}\}$ for each algorithm on SQuAD v1 and v2 dataset. Refer to Table 8 for detailed learning rate settings. The rank initialization and the number of trainable parameters for different algorithms are presented in Table 9. For PF2LoRA, we initialize the rank of client-specific adapter $\tilde{r}_k = \frac{r_k}{2} = 4$, and we set the best value of $r_{min} = 6, r_{max} = 14$ for HETLoRA. In addition, HETLoRA uses the sparsity parameter $\gamma = 0.99$ and the penalty factor $\lambda = 5.0 \times 10^{-3}$ on both SQuAD v1 and v2 datasets.

Table 7: The summary of GLUE benchmark.

| Corpus | # Train | # Test | # Lable | Metrics |
|--------|---------|--------|---------|---------|
| CoLA | 8.5k | 1k | 2 | Matthew's correlation |
| MNLI | 393k | 20k | 3 | Accuracy |
| SST-2 | 67k | 1.8k | 2 | Accuracy |
| QQP | 364k | 391k | 2 | Accuracy |
| QNLI | 108k | 5.7k | 2 | Accuracy |

Table 8: Learning rate choices for question-answering dataset SQuAD v1/v2.

| Method | SQuAD v1 | SQuAD v2 |
|--------|----------|----------|
| Centralized LoRA | $1.0 \times 10^{-3}$ | $5.0 \times 10^{-4}$ |
| HOMLoRA | $1.0 \times 10^{-3}$ | $5.0 \times 10^{-4}$ |
| Per-FedAvg-LoRA | $2.0 \times 10^{-3}/1.0 \times 10^{-3}$ | $1.0 \times 10^{-3}/1.0 \times 10^{-3}$ |
| HETLoRA | $5.0 \times 10^{-3}$ | $5.0 \times 10^{-3}$ |
| PF2LoRA | $2.0 \times 10^{-3}/1.0 \times 10^{-2}$ | $1.0 \times 10^{-3}/1.0 \times 10^{-2}$ |

### F.3 GPT-2 ON WEBNLG AND E2E NLG CHALLENGES

The optimal learning rates for each algorithm on WebNLG and E2E NLG Challenges are turned from the range $\{1.0 \times 10^{-4}, 5.0 \times 10^{-4}, 1.0 \times 10^{-3}, 2.0 \times 10^{-3}, 3.0 \times 10^{-3}, 4.0 \times 10^{-3}, 5.0 \times 10^{-3}\}$, and the learning rate settings are summarized in Table 10. For the rank initialization, we follow LoRA paper (Hu et al., 2021) and choose a small rank $r_k = 4$ for Centralized LoRA, HOMLoRA, and Per-FedAvg-LoRA. We turn the the best parameters and set $r_{min} = 6, r_{max} = 12$ for HETLoRA. PF2LoRA uses the same $r_k = 4$ for the common adapter and $\tilde{r}_k = 2$ for the client-specific adapter. The detailed rank settings and the number of trainable parameters are shown in Table 12. HETLoRA sets the sparsity parameter $\gamma = 0.99$ and the penalty factor $\lambda = 5.0 \times 10^{-4}$ on both WebNLG and E2E NLG Challenge datasets.

## G SUPPLEMENTARY EXPERIMENTAL RESULTS FOR TEXT CLASSIFICATION

This section provides experimental results about RoBERTa large model on GLUE benchmark. The comparison results with other baselines are shown in Table 13. We can observe that PF2LoRA achieves higher classification performance. For example, PF2LoRA outperforms HETLoRA by 3.88%, 22.24%, 2.99%, 13.89% and 2.69% on the five datasets, respectively.

## H SUPPLEMENTARY EXPERIMENTAL RESULTS FOR E2E NLG CHALLENGE

This section provides experimental results for E2E NLG dataset in Table 14. Compared to other federated baselines, our approach demonstrates the best performance on four metrics (BLEU, NIST, ROUGE-L, CIDEr) of five.

Table 9: Rank initialization and trainable parameters for DeBERTa v3.

| Method | Rank initialization | # Trainable parameters |
|--------|---------------------|------------------------|
| Centralized LoRA | $r_k = 8$ | 0.30M |
| HOMLoRA | $r_k = 8$ | 0.30M |
| Per-FedAvg-LoRA | $r_k = 8$ | 0.30M |
| HETLoRA | $r_{min} = 6, r_{max} = 14$ | 0.30M |
| PF2LoRA | $r_k = 8, \tilde{r}_k = 4$ | 0.44M |

Table 10: Learning rate choices for GPT-2 medium on NLG dataset WebNLG and E2E NLG Challenge.

| Method | WebNLG | E2E NLG Challenge |
|---|---|---|
| Centralized LoRA | $1.0 \times 10^{-3}$ | $1.0 \times 10^{-3}$ |
| HOMLoRA | $1.0 \times 10^{-3}$ | $1.0 \times 10^{-3}$ |
| Per-FedAvg-LoRA | $2.0 \times 10^{-3}/1.0 \times 10^{-4}$ | $2.0 \times 10^{-3}/2.0 \times 10^{-3}$ |
| HETLoRA | $2.0 \times 10^{-3}$ | $2.0 \times 10^{-3}$ |
| PF2LoRA | $2.0 \times 10^{-3}/1.0 \times 10^{-3}$ | $3.0 \times 10^{-3}/5.0 \times 10^{-4}$ |

Table 11: Learning rate choices for GPT2-XL on NLG dataset WebNLG.

| Method | WebNLG |
|---|---|
| HOMLoRA | $1.0 \times 10^{-3}$ |
| Per-FedAvg-LoRA | $1.0 \times 10^{-3}/1.0 \times 10^{-4}$ |
| HETLoRA | $1.0 \times 10^{-3}$ |
| PF2LoRA | $1.0 \times 10^{-3}/1.0 \times 10^{-4}$ |

Table 12: Rank initialization and trainable parameters for GPT-2.

| Method | Rank initialization | # Trainable parameters |
|---|---|---|
| Centralized LoRA | $r_k = 4$ | 0.39M |
| HOMLoRA | $r_k = 4$ | 0.39M |
| Per-FedAvg-LoRA | $r_k = 4$ | 0.39M |
| HETLoRA | $r_{min} = 6, r_{max} = 12$ | 0.81M |
| PF2LoRA | $r_k = 4, \tilde{r}_k = 2$ | 0.59M |

# I ABLATION STUDIES

We execute the ablation studies to explore (1) the performance comparison when other baselines have more trainable parameters than ours. (2) the impact of data heterogeneity on PF2LoRA and baselines. (3) the importance of bilevel optimization in our framework.

**Baselines with More Trainable Parameters.** The lightweight client-specific adapters introduce additional trainable parameters. For fair comparison with other baselines, we consider to increase the number of trainable parameters in other baselines. Specifically, we increase the initial rank $r_k$ (from 8 to 12) for baselines HOMLoRA and Per-FedAvg-LoRA in the text classification experiments. Note that HETLoRA has different rank initialization $r_{min} \leq r_k \leq r_{max}$ for different client $k$, so we count the average trainable parameters of the clients. we can also control the number of trainable parameters by specifying $r_{min}$ and $r_{max}$. We specify $r_{min} = 5, r_{max} = 12$ in CoLA dataset and $r_{min} = 8, r_{max} = 12$ in other four text classification datasets. The number of trainable parameters of each baseline and the corresponding test score in each dataset are summarized in Table 4. Even if other algorithms have more trainable parameters than our method, PF2LoRA still demonstrates the best performance. PF2LoRA, with negligible additional trainable parameters, significantly improves the performance in personalized federated learning.

## I.1 THE IMPACT OF HETEROGENEITY LEVELS

Heterogeneity level is regarded as an important factor in federated learning. In this section, we explore the impact of various heterogeneity levels on the performance of algorithms. We run PF2LoRA and other baselines on text classification datasets SST-2 and QNLI with three different heterogeneity levels $s = 0.6, 0.9, 1.0$. The accuracy results are shown in Table 16. PF2LoRA performs consistently well on different heterogeneity levels, and HETLoRA follows. The performance of HOMLoRA and Per-FedAvg-LoRA decreases significantly as the heterogeneity level increases. Especially, PF2LoRA outperforms other baselines in a large margin in the case of very high heterogeneity, e.g., $4.35\%$ higher than HETLoRA and $13.87\%$ higher than HOMLoRA on SST-2 dataset.

Table 13: Roberta-large results on GLUE benchmark. We report "Matthew's correlation" for CoLA and "Accuracy" for MNLI, SST-2, QQP and QNLI. Higher value means "better performance".

| Method | CoLA | MNLI | SST-2 | QQP | QNLI |
|---|---|---|---|---|---|
| Centralized LoRA | 57.32 | 84.71 | 93.67 | 88.43 | 90.27 |
| HOMLoRA | 51.71 | 74.51 | 93.33 | 79.76 | 89.63 |
| Per-FedAvg-LoRA | 51.20 | 75.68 | 92.64 | 81.83 | 79.49 |
| HETLoRA | 54.15 | 76.38 | 94.53 | 82.55 | 92.31 |
| PF2LoRA | **56.25** | **93.37** | **97.36** | **94.02** | **94.79** |

Table 14: GPT-2 generation results on E2E dataset.

| method | BLEU ↑ | NIST ↑ | MET ↑ | ROUGE-L ↑ | CIDEr ↑ |
|---|---|---|---|---|---|
| Centralized LoRA | 0.6833 | 8.5321 | 0.4642 | 0.7046 | 2.4023 |
| HOMLoRA | 0.5585 | 7.0986 | **0.4349** | 0.6095 | 1.8327 |
| Per-FedAvg-LoRA | 0.5683 | 7.1190 | 0.4327 | 0.6109 | 1.8984 |
| HETLoRA | 0.5505 | 7.0088 | 0.4093 | 0.5697 | 1.7167 |
| PF2LoRA | **0.5717** | **7.1621** | 0.4321 | **0.6111** | **1.9088** |

Table 15: GPT2-XL generation results on WebNLG dataset.

| Method | BLEU ↑ | MET ↑ | TER ↓ | ROUGE-L ↑ |
|---|---|---|---|---|
| HOMLoRA | 0.5768 | 0.7771 | **0.6103** | 0.3967 |
| Per-FedAvg-LoRA | 0.5783 | 0.7783 | 0.6157 | 0.3972 |
| HETLoRA | 0.5763 | 0.7789 | 0.6164 | 0.3922 |
| PF2LoRA | **0.5881** | **0.7832** | 0.6198 | **0.3978** |

Next, we further study the impact of relatively lower heterogeneity levels on the algorithms. We run PF2LoRA and other federated baselines on CoLA dataset in the heterogeneity levels of $s = 0.2$, $s = 0.3$ and $s = 0.4$, and the results of "Matthew's correlation" are summarized in Table 17. PF2LoRA outperforms all the baselines consistently in various heterogeneity levels. For example, PF2LoRA surpasses the best baseline HETLoRA by $4.36\%$, $0.8\%$ and $12.15\%$ in heterogeneity levels of $s = 0.2, s = 0.3, s = 0.4$ respectively. Therefore, our algorithm PF2LoRA demonstrates the high robustness to heterogeneity levels.

## I.2 PERFORMANCE WITH/WITHOUT BILEVEL OPTIMIZATION

We conduct an ablation study to verify the effect of bilevel optimization. Instead of applying bilevel optimization in (3), we update parameters in the common and client-specific adapters simultaneously.

$$\min_{x, y_k} \frac{1}{M} \sum_{k=1}^{M} f_k(x, y_k),$$
$$f_k(x, y_k) := \mathbb{E}_{\xi \sim \mathcal{D}_k} F_k(x, y_k; \xi), \tag{22}$$

where $\mathcal{D}_k$ is the data on client $k$. Specifically, we keep the optimizer settings mentioned in Section 6.1.1, where a SGD optimizer is applied to updating the client-specific adapter and an AdamW optimizer to the common adapter. The difference is that we do not use the hypergradient (4) to update the common adapter, instead update it by $x_k^{t+1} = x_k^t - \eta \nabla_x F_k(x_k^t, y_k^t; \xi_k^t)$. We execute our "two-level low rank adaptation" framework without bilevel optimization on text classification of GLUE benchmark. For fair comparison, we keep the same hyperparameter settings as that in Section 6.1.1, including heterogeneity level, learning rates, communication rounds, communication interval and initial rank dimension on the same dataset. The comparison results are shown in Figure 3, where we can see that the framework with bilevel optimization (BO) always performs better than that without BO, especially on harder classification task, such as CoLA dataset.

Table 16: Results in different heterogeneity levels. We use "Accuracy" to measure the performance here, and higher value means "better performance".

| Methods | SST-2 | | | QNLI | | |
|---|---|---|---|---|---|---|
| | s=0.6 | s=0.9 | s=1.0 | s=0.6 | s=0.9 | s=1.0 |
| HOMLoRA | 92.66 | 92.47 | 83.49 | 86.62 | 85.45 | 67.32 |
| Per-FedAvg-LoRA | 90.80 | 90.56 | 85.29 | 85.32 | 78.59 | 50.48 |
| HETLoRA | 93.74 | 93.67 | 91.11 | 89.28 | 91.86 | 89.09 |
| PF2LoRA | **94.12** | **95.85** | **95.07** | **92.87** | **94.18** | **93.64** |

Table 17: Matthew's correlation on CoLA in different heterogeneity levels. Higher value means "better performance".

| Methods | CoLA | | |
|---|---|---|---|
| | s=0.2 | s=0.3 | s=0.4 |
| HOMLoRA | 52.91 | 50.75 | 43.17 |
| Per-FedAvg-LoRA | 53.48 | 51.11 | 44.44 |
| HETLoRA | 53.86 | 53.76 | 45.03 |
| PF2LoRA | **56.20** | **54.19** | **50.50** |

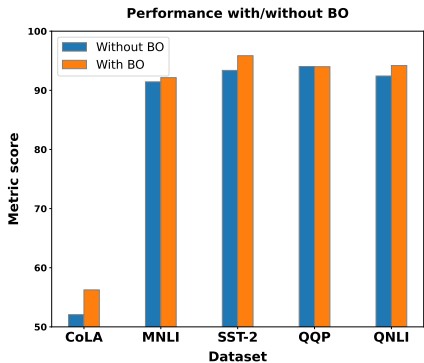

Figure 3: Performance comparison with/without bilevel optimization (BO). We report "Matthew's correlation" for CoLA and "Accuracy" for MNLI, SST-2, QQP and QNLI. Higher score means "better performance"

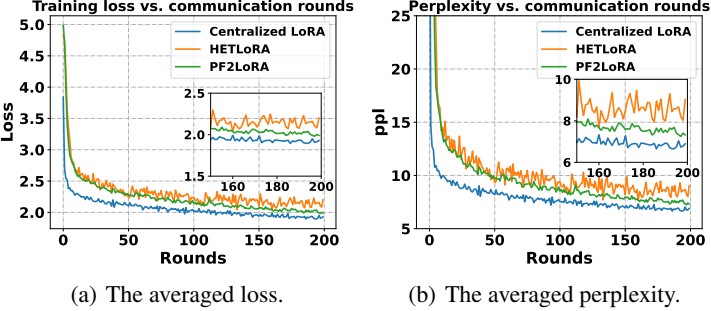

(a) The averaged loss.          (b) The averaged perplexity.

Figure 4: The averaged training loss and perplexity on natural language generation task of WebNLG.

## J   STABILITY ANALYSIS

Despite that HETLoRA is a strong baseline which performs usually well on heterogeneous data. However, we empirically observe that the training process of HETLoRA is not as stable as ours and Centralized LoRA in Figure 4, where the training loss and perplexity (ppl) are averaged across

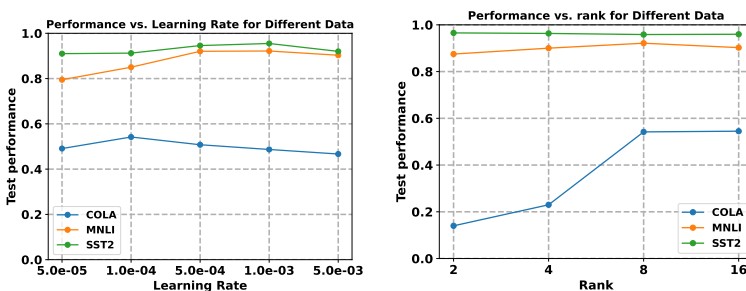

(a) PF2LoRA performance vs. learning rate $\alpha$.    (b) PF2LoRA performance vs. rank $r$.

Figure 5: Sensitivity analysis of hyperparameters.

all the clients. A possible and reasonable explanation is that HETLoRA adopts dynamical rank pruning and matrices truncation which directly change the intrinsic structure of local adapters, leading to unstable training. On the one hand, pruning removes some columns or rows from the original weights, which can degrade the model performance and require some steps of fine-tuning to recover the performance (Han et al., 2015). On the other hand, each client is required to truncate the common adapter matrices to align the matrices' dimensions at each communication round, which inevitably loses some potentially important information. In contrast, our method circumvents the alignment issue of adapter matrices by assigning a uniform rank $r_k$ to the common adapter and uniform $\tilde{r}_k$ to all the client-specific adapters.

## K  SENSITIVITY ANALYSIS OF HYPERPARAMETER

We run our algorithm PF2LoRA on GLUE benchmarks using a hyperparameter sweep, and the results are presented in Figure 5. In our setting, we require the local adapter to be light-weight, so the rank of local adapters is always small, i.e., $\tilde{r} = 2$. We perform a hyperparameter sweep on the local learning rate $\alpha$ and the rank of the common adapter, respectively. As you see in subfigure 2(a), our algorithm is pretty robust to the learning rate $\alpha$. Since COLA dataset is more challenging than others, a larger rank is helpful to improve the model performance, but the performance keeps almost the same when the rank is larger than 8. Our algorithm also exhibits high robustness on data MNLI and SST-2.

## L  COMPUTATION AND COMMUNICATION COST

We evaluated the total computational costs (FLOPs) and communication costs in a single communication round for each algorithm on GLUE benchmark. The results are summarized in Table 18. The computational cost (FLOPs) per round are determined by the number of model parameters and the forward/backward propagation operations. As PF2LoRA requires to compute the hessian-vector product for hypergradient estimation, it incurs a higher computational cost. But the communication cost of PF2LoRA remains consistent with that of HOMLoRA and Centralized LoR, as the communication parameters in PF2LoRA are only global adapters that have the same rank $r_k = 8$ with that in HOMLoRA and Centralized LoRA. Instead, HETLoRA has a higher parameter rank requirement for a high performance, resulting in increased communication costs.

Table 18: Computational/Communication costs per communication round.

| Method | TFLOPs/round | Communication parameters/round |
|---|---|---|
| Centralized LoRA ($r_k = 8$) | 258.40 | 0.30M |
| HOMLoRA ($r_k = 8$) | 258.40 | 0.30M |
| Per-FedAvg-LoRA ($r_k = 8$) | 908.00 | 0.30M |
| HETLoRA ($r_{max} = 12, r_{min} = 8$) | 272.60 | 0.35M |
| PF2LoRA ($r_k = 8, \tilde{r} = 2$) | 1202.40 | 0.30M |

## M    Generated Result of NLU

### M.1    Generated Examples for E2E NLG Challenge

Table 19 and 20 show the generated examples of algorithm HETLoRA and PF2LoRA. The federated fine-tuning experiments are run across 8 clients on E2E NLG Challenges, where we construct the heterogeneous data by the "name" of restaurants, thus each client has different meta-information from different restaurants. There are 18 restaurants in the test set distributed in 8 clients. We show the generated examples based given context information on each client, while multiple references are provided to evaluate the quality of generated contents. We compare the generated contents from HETLoRA and PF2LoRA. In most cases, PF2LoRA can generate more complete and logically coherent sentences. For example, the generated contents on client 4 and client 7, HETLoRA misses some important information (highlighted in green). The examples on client 1, 2, 3 and 4, PF2LoRA produces more grammatically coherent sentences than HELoRA.

### M.2    Generated Examples for WebNLG

For WebNLG dataset, we construct the heterogeneity data by the topics ['Airport', 'Astronaut', 'Building', 'City', 'ComicsCharacter', 'Food', 'Monument', 'SportsTeam', 'University', 'WrittenWork']. These topics are distributed across 8 clients. Thus, the language style varies with the text topics. We run the personalized federated fine-tuning across 8 clients and report the generated examples for given test context. The comparison results show that PF2LoRA can generate more complete and high quality sentences than HETLoRA. For example on client 0 and 1, HETLoRA misses key words "runwayname", "test pilot", which actually are important information. On client 2 and 5, HETLoRA generates incorrect information, while PF2LoRA produces accurate sentences. Refer to Table 21 for comparison details.

## N    The Use of Large Language Models (LLMs)

LLMs are not involved in our research methodology or analysis. Their use is limited to polish the writing.

Table 19: The generated examples for E2E NLG Challenges

| | Client 0 |
|---|---|
| Context | name : blue spice \| type : pub \| food : english \| area : riverside \| family friendly : yes \| near : rainbow vegetarian café |
| References | in riverside , near the rainbow vegetarian café , you can find a family friendly pub called blue spice . |
| | if you like english food there is a family - friendly pub called blue spice near the rainbow vegetarian café in riverside . |
| | the blue spice is a child - friendly , english pub located in riverside area , near rainbow vegetarian café . |
| | blue spice is located near rainbow vegetarian café in the riverside area and is a kid friendly pub that serves english food . |
| | there is a pub called blue spice which serves english food , is kid friendly , and is in riverside near rainbow vegetarian café . |
| | blue spice is a child - friendly pub near rainbow vegetarian café in the riverside area . |
| | blue spice near rainbow vegetarian café in riverside is a pub serving english meals and child friendly |
| | the blue spice is a pub . it is located near rainbow vegetarian café in the area of riverside . this is a family friendly pub |
| | serving english food . |
| | an english serving child friendly pub in riverside is blue spice near rainbow vegetarian café |
| | there is a pub that provides food and is children friendly , near rainbow vegetarian café and the riverside and is called blue spice . |
| | situated near the rainbow vegetarian café in the riverside area of the city , the blue spice pub , is ideal if you fancy traditional english food whilst out with the kids . |
| HETLoRA | blue spice is a pub near rainbow vegetarian café in the riverside area . *it is family friendly and serves english food* . |
| PF2LoRA | *blue spice is a family friendly pub that serves english food* . it is located in the riverside area near the rainbow vegetarian café . |

| | Client 1 |
|---|---|
| Context | name : the cricketers \| type : coffee shop \| customer rating : low \| family friendly : no \| near : ranch |
| References | the cricketers is a coffee shop with a low customer rating , located near ranch . it is not family - friendly . |
| HETLoRA | *city centre coffee shop , the cricketers* , is not family - friendly and has a low customer rating . it is located near ranch . |
| PF2LoRA | north of ranch , *there is a coffee shop called the cricketers* . it is not family - friendly and has a low customer rating . |

| | Client 2 |
|---|---|
| Context | name : the mill \| type : restaurant \| food : english \| price : moderate \| customer rating : 3 out of 5 \| area : riverside \| family friendly : yes \| near : café rouge |
| References | the riverside area has restaurant near the café rouge that is both in the moderate price range and kid friendly called the mill . it has a 3 out of 5 customer rating and serves english food . |
| | the riverside area near café rouge has a restaurant that is kids - friendly . it has a price range in the mill . i give the food a 3 out of 5 . |
| | the mill is a kids friendly restaurant that has moderate prices and serves english food . it has a 3 out of 5 customer rating and is located in the riverside area near the café rouge . |
| HETLoRA | the mill is a moderately priced english restaurant *near café rouge in the riverside area* . it is kid friendly and has a customer rating of 3 out of 5 . |
| PF2LoRA | the mill is a moderately priced restaurant *in the riverside area near café rouge* . it serves english food and is kid friendly . it has a customer rating of 3 out of 5 . |

| | Client 3 |
|---|---|
| Context | name : the phoenix \| type : pub \| food : french \| price : £ 20 - 25 \| customer rating : high \| area : riverside \| family friendly : no \| near : crowne plaza hotel |
| References | a pub that is not kid friendly is located in the riverside area near crowne plaza hotel . it is named the phoenix , has french food and price range of £ 20 - £ 30 and a high customer rating . |
| | the phoenix , which is a pub that is not kid friendly , is near crowne plaza hotel and serves french food in the price range of £ 20 - 25 in the riverside area . it has a high customer rating . |
| HETLoRA | the phoenix is a pub *near the crowne plaza hotel in the riverside area* . it has a high customer rating and a price range of £ 20 - 25 . it is not kid friendly . |
| PF2LoRA | the phoenix is a pub *in the riverside area near the crowne plaza hotel* . it serves french food with a price range of £ 20 - 25 and has a high customer rating . it is not kid friendly . |

Table 20: The generated examples for E2E NLG Challenges (continued).

| | Client 4 |
|---|---|
| Context | name : the punter \| type : restaurant \| food : italian \| price : cheap \| customer rating : average \| area : riverside \| family friendly : no \| near : rainbow vegetarian café |
| References | hello and welcome to the punter , we serve the finest italian food around and have an average customer rating this is very good for a restaurant we are near rainbow vegetarian café and our area is the riverside our price range is very cheap for such good food at the moment we are not family - friendly . a restaurant serving italian food for adults can be found on the riverside near rainbow vegetarian café . the punter has average ratings , and cheap prices |
| HETLoRA | the italian restaurant the punter is located in the riverside area near rainbow vegetarian café . it is not family - friendly and has an average customer rating . |
| PF2LoRA | the punter is a cheap italian restaurant near the rainbow vegetarian café in the riverside area . it has an average customer rating and is not family - friendly . |

| | Client 5 |
|---|---|
| Context | name : the vaults \| type : pub \| food : japanese \| price : less than £ 20 \| customer rating : average \| area : city centre \| family friendly : no \| near : raja indian cuisine |
| References | no , the vaults is fast food with pub on side raja indian cuisine in the city centre with price less than £ 20 average the vaults pub that is less than 20 pounds and has an average customer rating and is family - friendly . they serve japanese food near the center of the city and also near the raja indian cuisine . the vaults is japanese with pub on side raja indian cuisine in the city centre with average no less than £ 20 |
| HETLoRA | the vaults is a pub that serves japanese food . it is located in the city centre near raja indian cuisine . it is not family - friendly and has a price range of less than £ 20 . it has an average customer rating . |
| PF2LoRA | the vaults is a japanese pub located in the city centre near raja indian cuisine . it has an average customer rating and a price range of less than £ 20 . it is not family - friendly . |

| | Client 6 |
|---|---|
| Context | name : the waterman \| type : pub \| food : italian \| price : high \| area : riverside \| family friendly : yes \| near : raja indian cuisine |
| References | the waterman is a high price range italian pub in the riverside area near raja indian cuisine . it is children friendly . near raja indian cuisine , the waterman pub has high prices and facilities for the family . the pub the waterman is an expensive family friendly establishment located near raja indian cuisine . located near raja indian cuisine in the riverside area , the waterman is a high priced children friendly pub serving italian food . conveniently located on the riverside , close to raja indian cuisine , the waterman is an elite , but family friendly established pub . |
| HETLoRA | the waterman is a high priced italian pub near raja indian cuisine in the riverside area . it is children friendly . |
| PF2LoRA | the waterman is an italian pub located in the riverside area near raja indian cuisine . it is children friendly and has a high price range . |

| | Client 7 |
|---|---|
| Context | name : name : wildwood \| type : pub \| food : indian \| area : city centre \| family friendly : yes \| near : raja indian cuisine |
| References | located near the city centre and raja indian cuisine the wildwood is a family friendly indian pub . wildwood is in the city centre area near raja indian cuisine . it is a pub that serves indian food and is family friendly . wildwood also offers indian food to go along with the family friendly pub located near raja indian cuisine |
| HETLoRA | aji indian cuisine pub in the city centre near raja indian cuisine is kid friendly and serves indian food . |
| PF2LoRA | a pub near raja indian cuisine in the city centre called wildwood serves indian food and is kid friendly . |

Table 21: The generated examples for WebNLG.

| | |
|---|---|
| **Client 0 (Airport)** | |
| Context | al_asad_airbase : operatingorganisation : united_states_air_force \| al_asad_airbase : runwaylength : 3992 . 88 \| al_asad_airbase : location : " al anbar province , iraq " \| al_asad_airbase : icao_location_identifier : " oraa " \| al_asad_airbase : runwayname : " 08 / 26 " |
| References | al asad air base has a runway name of 08 / 26 which is 3992 . 8 in length . it is situated in the al anbar province of iraq , is operated by the united states air force and has the icao location identifier oraa . the united states airport operates the al asad airbase which is located in the al anbar province , iraq . the icao location identifer of al asad airbase is oraa and the length is 3992 . 88m and the runway is known as 08 / 26 . |
| HETLoRA | ! the united states air force is the operating organisation for al asad airbase which is located in al anbar province , iraq . the airbase has a runway length of 3992 . 88 and the icao location identifier is oraa . |
| PF2LoRA | the united states air force is the operating organisation for al asad airbase in al anbar province , iraq . the icao location identifier of al asad airbase is oraa and it has a runway length of 3992 . 88 . the runway name of the airbase is 08 / 26 . |
| **Client 1 (Astronaut)** | |
| Context | alan_shepard : status : " deceased " \| alan_shepard : almamater : " nwc , m . a . 1957 " \| alan_shepard : deathplace : california \| alan_shepard : occupation : test_pilot \| alan_shepard : birthplace : new_hampshire \| alan_shepard : was selected by nasa : 1959 \| alan_shepard : birthdate : " 1923 - 11 - 18 " |
| References | alan shepard has died in california . he was born on 18 november 1923 in new hampshire and attended school at nwc , graduating in 1957 with an ma . he became a test pilot and was eventually selected by nasa in 1959 . alan shepard was born in new hampshire on november 18th , 1923 . he graduated from nwc in 1957 with an m . a . he was selected by nasa in 1959 and he was a test pilot . he died in california . alan shepard , born on november 18 , 1923 , graduated from nwc in 1957 with an m . a . alan shepard served as a test pilot , and was selected by nasa in 1959 . alan shepard , born in new hampshire , died in california , . |
| HETLoRA | alan shepard was born on november 18th , 1923 in new hampshire . he graduated from nwc in 1957 with an m . a . and was selected by nasa in 1959 . he died in california . |
| PF2LoRA | alan shepard was born in new hampshire on november 18th , 1923 . he graduated from nwc with a m . a . in 1957 . he was selected by nasa in 1959 and served as a test pilot . alan shepard died in california . |
| **Client 2 (Building)** | |
| Context | adisham_hall : country : sri_lanka \| sri_lanka : capital : sri_jayawardenepura_kotte \| sri_lanka : currency : sri_lankan_rupee |
| References | sri jayawardenepura kotte is the capital of sri lanka , which uses the sri lankan rupee as its currency and is the location of adisham hall . sri jayawardenepura kotte is the capital of sri lanka , whose currency is the rupee . adisham hall is located in sri lanka . |
| HETLoRA | college adisham hall is located in the country of sri lanka , where the capital is sri jayawardenepura kotte and the currency is the sri lankan rupee . |
| PF2LoRA | alan adisham hall is located in sri lanka , the capital of which is sri jayawardenepura kotte . the currency of sri lanka is the sri lankan rupee . |
| **Client 3 (File)** | |
| Context | big_hero_6_ ( film ) : starring : ryan_potter \| big_hero_6_ ( film ) : distributor : walt_disney_studios_motion_pictures \| baymax : series : big_hero_6_ ( film ) |
| References | the movie big hero 6 stars ryan potter which has baymax as one of its characters , was distributed by walt disney studios motion pictures . baymax is a character in the big hero 6 film starring ryan potter and distributed by walt disney studios motion pictures . walt disney studio motion picture distributed the film big hero 6 , in which ryan potter starred and baymax is a character . |
| HETLoRA | ! baymax is a character in the film big hero 6 which stars ryan potter . the film was distributed by walt disney studios motion pictures . |
| PF2LoRA | walt disney studios motion pictures is the distributor of big hero 6 , a film in which baymax is a character . the film stars ryan potter . |

Table 22: The generated examples for WebNLG (continued).

| | Client 4 (Food) |
|---|---|
| Context | bacon_sandwich : dishvariation : blt \| bacon_sandwich : mainingredients : " bread and bacon , with a condiment , often ketchup or brown sauce " \| bacon_sandwich : country : united_kingdom \| bacon_sandwich : ingredient : ketchup \| bacon_sandwich : alternativename : " bacon butty , bacon sarnie , rasher sandwich , bacon sanger , piece ' n bacon , bacon cob , bacon barm , bacon muffin " |
| References | the bacon sandwich , also known as : bacon butty , bacon sarnie , rasher sandwich , bacon sanger , piece n ' bacon , bacon cob , bacon barm , blt and bacon muffin , is from the united kingdom . it includes bread , bacon and ketchup or brown sauce . the variant blt bacon sandwich includes : bread , bacon , and ketchup or brown sauce . it originates from the united kingdom and goes by the various names : bacon butty , bacon sarnie , rasher sandwich , bacon sanger , piece n ' bacon , bacon barm , bacon cob and bacon muffin . |
| HETLoRA | the bacon sandwich is a variation of the blt . the main ingredients are bread and bacon , with a condiment , often ketchup or brown sauce . it is also known as a bacon butty , bacon sarnie , rasher sandwich , bacon sanger , piece ' n bacon , bacon cob , bacon barm , bacon muffin . |
| PF2LoRA | the bacon sandwich is a variation of the blt and is a dish from the united kingdom . the main ingredients of a bacon sandwich are bread and bacon with ketchup or brown sauce as a condiment . the bacon sandwich can also be known as a bacon butty , bacon sarnie , rasher sandwich , bacon sanger , piece ' n bacon , bacon cob , bacon barm , or bacon muffin . |

| | Client 5 (SportsTeam) |
|---|---|
| Context | a . f . c . _blackpool : manager : stuart_parker_ ( footballer ) \| stuart_parker_ ( footballer ) : club : chesterfield_f . c . \| stuart_parker_ ( footballer ) : club : stockport_county_f . c . \| a . f . c . _blackpool : ground : blackpool \| blackpool : leader : labour_party_ ( uk ) |
| References | a . f . c . blackpool is in blackpool , which council is labour , it has had stuart parker as their manager , whose football club was stockport county f . c and is attached to chesterfield football club . alan shepard , born on november 18 , 1923 , graduated from nwc in 1957 with an m . a . alan shepard served as a test pilot , and was selected by nasa in 1959 . alan shepard , born in new hampshire , died in california , . |
| HETLoRA | ! stuart parker ( footballer ) is the manager of a . f . c . blackpool who play in blackpool , where the leader is the labour party ( uk ) and the ground is called blackpool . |
| PF2LoRA | ! a . f . c . blackpool is in blackpool , where the leader is the labour party ( uk ) . the club is managed by stuart parker ( footballer ) who played for chesterfield fc and stockport county f . c . |

| | Client 6 (University) |
|---|---|
| Context | romania : ethnicgroup : germans_of_romania \| romania : leadertitle : prime_minister_of_romania \| alba_iulia : country : romania \| romania : leadername : klaus_iohannis \| romania : capital : bucharest \| 1_decembrie_1918_university : city : alba_iulia \| romania : anthem : deșteaptă - te , _române ! |
| References | the 1 decembrie 1918 university is in the city alba iulia in romania . klaus iohannis the leader of romania and they also have a prime minister . the germans of romania are the main ethnic group in romania and the capital is bucharest . the romania anthem is deșteaptă - te , române ! |
| HETLoRA | ! the 1 decembrie 1918 university is located in alba iulia , romania . the country ' s leader is prime minister klaus iohannis and its capital is bucharest . the anthem of the country is deșteaptă - te , române ! |
| PF2LoRA | the 1 decembrie 1918 university is located in alba iulia , romania . romania ' s capital is bucharest and its leader is prime minister klaus iohannis . the national anthem of romania is deșteaptă - te , române ! and its ethnic group is the germans of romania . |

| | Client 7 (WrittenWork) |
|---|---|
| Context | administrative_science_quarterly : publisher : cornell_university \| cornell_university : affiliation : association_of_public_and_land - grant_universities \| cornell_university : affiliation : association_of_american_universities \| cornell_university : president : elizabeth_garrett \| cornell_university : city : ithaca , _new_york |
| References | administrative science quarterly was published by cornell university , located in ithaca , new york , and affiliated with the association of public and land grant universities , as well as with the association of american universities . president of cornell university is elizabeth garrett . |
| HETLoRA | ! the administrative science quarterly is published by cornell university , which is affiliated with the association of public and land grant universities and the association of american universities . it is located in ithaca , new york . the president of cornell university is elizabeth garrett . |
| PF2LoRA | the administrative science quarterly is published by cornell university , ithaca , new york . the university is affiliated with the association of public and land grant universities and the association of american universities . the president of the university is elizabeth garrett . |

