# OpenReview forum: "Personalized Federated Fine-tuning for Heterogeneous Data: An Automatic Rank Learning Approach via Two-Level LoRA"
_ICLR.cc/2026/Conference — Submitted to ICLR 2026_

### Official Review · Reviewer_YmR9 · 2025-10-27

**Soundness:** 3
**Presentation:** 2
**Contribution:** 2
**Rating:** 4
**Confidence:** 4

**Summary:**

This work proposes PF2LoRA, a personalized federated fine-tuning algorithm built on an automatic rank learning approach via two-level
LoRA. Given the pretrained language model whose weight is frozen, their algorithm aims to learn two levels of adaptation simultaneously: the first level aims to learn a common adapter for all clients, while the second level fosters individual client personalization.

**Strengths:**

- This work proposes a novel bilevel formulation for personalized fine-tuning on heterogeneous data and develops a two-level low-rank adaptation framework to efficiently fine-tune the foundation model in the scenario of FL.
- The proposed PF2LoRA has the ability to adaptively determine a suitable rank based on an individual client’s data, rather than relying on a predefined rank that is agnostic to data heterogeneity.

**Weaknesses:**

- The concepts of global adapters and client-specific adapters have been widely adopted in FL with LoRA fine-tuning [1-3], which diminishes the novelty of this work.
- The explanation of how the algorithm can achieve dynamic rank is not well articulated. Specifically, in Lines 198-199, "our specific parameterization (2) explicitly encourages each adapter \\( W_k \\) for the k-th client to vary over k: it can have different ranks in the range \\( (r - \widetilde{r}, r + \widetilde{r}) \\)." How is this achieved? Why can the rank be between \\( (r - \widetilde{r}, r + \widetilde{r}) \\)?
- The theoretical analysis is unrelated to the proposed method. The theoretical analysis is based on the single machine case (M = 1), which is unrelated to the proposed method in the FL setting.
- Many works on FL with LoRA fine-tuning across heterogeneous ranks [4-10] are missing and need to be compared.

[1] Long, G., Shen, T., Jiang, J. and Blumenstein, M., 2024. Dual-personalizing adapter for federated foundation models. Advances in Neural Information Processing Systems, 37, pp.39409-39433. \
[2] Qi, J., Luan, Z., Huang, S., Fung, C., Yang, H. and Qian, D., 2024. Fdlora: Personalized federated learning of large language model via dual lora tuning. arXiv preprint arXiv:2406.07925. \
[3] Nguyen, D.P., Munoz, J.P., Roosta, T. and Jannesari, A., 2025, May. Federated Multimodal Learning with Dual Adapters and Selective Pruning for Communication and Computational Efficiency. In 2025 IEEE 25th International Symposium on Cluster, Cloud and Internet Computing (CCGrid) (pp. 01-10). IEEE. \
[4] Byun, Y. and Lee, J., 2024. Towards federated low-rank adaptation of language models with rank heterogeneity. arXiv preprint arXiv:2406.17477. \
[5] Chen, S., Tavallaie, O., Nazemi, N. and Zomaya, A.Y., 2024, November. Rbla: Rank-based-lora-aggregation for fine-tuning heterogeneous models in flaas. In International Conference on Web Services (pp. 47-62). Cham: Springer Nature Switzerland. \
[6] Su, Y., Yan, N., Deng, Y., Dohler, M. and Schober, R., 2024. HAFLQ: Heterogeneous Adaptive Federated LoRA Fine-tuned LLM with Quantization. arXiv preprint arXiv:2411.06581. \
[7] Bai, J., Chen, D., Qian, B., Yao, L. and Li, Y., 2024. Federated fine-tuning of large language models under heterogeneous tasks and client resources. Advances in Neural Information Processing Systems, 37, pp.14457-14483. \
[8] Chen, S., Tavallaie, O., Nazemi, N., Chen, X. and Zomaya, A.Y., 2024. Autorank: Mcda based rank personalization for lora-enabled distributed learning. arXiv preprint arXiv:2412.15553. \
[9] Wu, F., Hu, J., Min, G. and Wang, S., 2025. Adaptive Rank Allocation for Federated Parameter-Efficient Fine-Tuning of Language Models. arXiv preprint arXiv:2501.14406. \
[10] Liu, Q., Zhang, Z., Yao, X. and Liu, B., 2025. HLoRA: Efficient federated learning system for LLM heterogeneous fine-tuning. arXiv preprint arXiv:2503.00813.

**Questions:**

Lines 240-242, "Empirically, we adopt AdamW as the upper-level optimizer (line 6) and SGD as the lower-level optimizer (line 5) to fine-tune a language model." Why choose these two optimizers? Is this based on some preliminary experimental results?

---

### Official Review · Reviewer_yZVy · 2025-10-31

**Soundness:** 3
**Presentation:** 3
**Contribution:** 2
**Rating:** 4
**Confidence:** 3

**Summary:**

This paper proposes PF2LoRA, a novel personalized federated fine-tuning algorithm for heterogeneous data using a two-level LoRA framework. The method introduces a bilevel optimization formulation where the first level learns a common adapter shared across clients, and the second level learns lightweight, client-specific adapters. A key contribution is the ability to automatically learn client-specific ranks based on local data heterogeneity, without relying on predefined rank bounds or extensive hyperparameter tuning. The authors demonstrate through synthetic and real-world experiments on NLU and NLG tasks that PF2LoRA outperforms existing federated fine-tuning baselines, including HETLoRA, with minimal additional memory overhead.

**Strengths:**

1. The two-level LoRA formulation and automatic rank learning mechanism are novel and creatively address data heterogeneity in federated fine-tuning.

2. The method is rigorously evaluated across multiple tasks and models, with thorough comparisons to state-of-the-art baselines.

3. The paper is well-written, with clear explanations of the motivation, method, and results.

4. The approach reduces the need for manual rank tuning and improves performance in personalized federated learning, which is a growing area of importance.

**Weaknesses:**

1. While the method is evaluated on language models, its applicability to other modalities (e.g., vision, multimodal) is not explored, though this is acknowledged as future work.

2. As shown in Table 18, PF2LoRA incurs the highest computational cost per communication round, representing a significant trade-off for its performance gains that may limit deployment on highly resource-constrained devices.

3. The theoretical analysis is limited to a single-machine setting; extending it to the full federated case would strengthen the theoretical contribution.

4.  While an improvement over HETLoRA, PF2LoRA still introduces new hyperparameters (e.g., the client-adapter rank $\tilde{r}$), whose selection remains heuristic and lacks a principled guidance strategy.

5. The ablation study shows a performance drop without bilevel optimization, but a more detailed analysis quantifying its contribution relative to the two-level adapter structure itself would strengthen the argument for its necessity.

**Questions:**

1. Could the authors discuss the scalability of PF2LoRA when the number of clients is large (e.g., >100)? How does the bilevel optimization scale in such settings?

2. The paper mentions that the client-specific adapters are not communicated. Have the authors considered the impact of client drift or non-IID data on the convergence of the common adapter?

3. In the ablation study, the framework without bilevel optimization still performs reasonably well. Can the authors further analyze the relative contribution of the bilevel formulation vs. the two-level adapter structure?

4. How sensitive is PF2LoRA to the choice of $\tilde{r}$ (e.g., $\frac{r}{2}$ or $\frac{r}{4}$)? Is there a principled way to set this parameter?

---

### Official Review · Reviewer_4oqs · 2025-10-31

**Soundness:** 2
**Presentation:** 3
**Contribution:** 2
**Rating:** 2
**Confidence:** 4

**Summary:**

This paper studies personalized federated fine-tuning for language models under heterogeneous client data. Existing federated LoRA approaches use fixed or independent low-rank adapters, which may not suit diverse client data. To address this, PF2LoRA is proposed, featuring a two-level LoRA design that simultaneously learns a common adapter for all clients and a personalized adapter for each client. PF2LoRA adaptively determines the optimal rank for each client based on local data, introducing minimal additional memory overhead. Experiments on natural language understanding and generation tasks show that PF2LoRA consistently outperforms existing federated fine-tuning methods.

**Strengths:**

1. The paper is clearly presented and easy to follow.
2. The paper is well motivated and the problem is clearly defined.
3. The paper provides a theoretical analysis.

**Weaknesses:**

1. The method lacks technical novelty. The proposed PF2LoRA method is largely similar to FedDPA [1], i.e., one LoRA shared across clients and another serving as a personalized adapter.  However, the author do not cite and discuss it. The author should provide a detailed discussion on the differences between the proposed approach and FedDPA.
[1] Dual-Personalizing Adapter for Federated Foundation Models. NeurIPS, 2024.

2.  The paper would benefit from a discussion of its limitations and potential directions for future work.

3. Experiments are limited to smaller models such as RoBERTa and GPT-2; evaluating on larger models (e.g., LlaMA2-7B and Qwen2.5-32B) would better demonstrate scalability.

**Questions:**

Please see weaknesses.

---

### Official Review · Reviewer_b8Qd · 2025-10-31

**Soundness:** 2
**Presentation:** 3
**Contribution:** 2
**Rating:** 4
**Confidence:** 4

**Summary:**

The contextual background of this study is an NLP task, which employs a federated learning paradigm to fine-tune foundation models using LoRA across multiple clients. The objective is to achieve personalized adaptation while minimizing computational overhead. To this end, this paper proposes PF2LoRA, which introduces a two-level LoRA structure that effectively decouples into a common adapter and a personal adapter, thereby enabling lightweight personalization. Experimental results demonstrate that PF2LoRA achieves certain performance improvements compared to both a homogeneous single-LoRA configuration and a state-of-the-art method from 2024.

**Strengths:**

1.This work presents solid research, supported by great experimental validation across multiple NLP tasks and datasets.
2.The theoretical foundation is rich, with detailed proofs provided in the Appendix.
3.The methodology is described with sufficient clarity, effectively conveying the core insight of the proposed approach.

**Weaknesses:**

1. The novelty of the proposed method is somewhat limited, as it constitutes a relatively straightforward application of the LoRA framework. This is illustrated by prior works such as:
[1]FedDecomp (MM 2024): This work also employs LoRA as a personalized adapter and similarly finds that preserving client-specific knowledge requires only lower-capacity personalized parameters, although it was applied to a computer vision (CV) task.
[2]PERADA (CVPR 2024): This study likewise designs separate personalized and global adapters, but also within the domain of CV.

2. The paper lacks a comprehensive analysis of computational and communication efficiency.

[1] Wu X, Liu X, Niu J, et al. Decoupling general and personalized knowledge in federated learning via additive and low-rank decomposition[C]//Proceedings of the 32nd ACM International Conference on Multimedia. 2024: 7172-7181.
[2]Xie C, Huang D A, Chu W, et al. Perada: Parameter-efficient federated learning personalization with generalization guarantees[C]//Proceedings of the IEEE/CVF conference on computer vision and pattern recognition. 2024: 23838-23848.

**Questions:**

1. Whether foundation models can even be deployed on client devices is a significant open question. Has the paper discussed the resources required for clients to perform fine-tuning tasks on such large foundation models?
2. Is it truly necessary to perform federated fine-tuning on a foundation model that has already been fully pre-trained? Is there any experimental evidence demonstrating that the original foundation model fails to handle personalized NLP tasks?
3. Why was prompt tuning not included as a baseline? What are its clear disadvantages compared to the LoRA approach within the federated learning paradigm? For the authors' reference, I am attaching a previously seen method that employs prompt tuning for federated fine-tuning:
[1] Bai S, Zhang J, Guo S, et al. Diprompt: Disentangled prompt tuning for multiple latent domain generalization in federated learning[C]//Proceedings of the IEEE/CVF Conference on Computer Vision and Pattern Recognition. 2024: 27284-27293.

---

### Meta-Review · Area_Chair_Gwub · 2026-01-04

**Summary:**

**Summary**: This paper introduces PF2LoRA, a personalized federated fine-tuning algorithm for language models operating with heterogeneous client data. It addresses the limitation of existing federated LoRA methods that often rely on predefined or fixed low-rank adapter ranks, which may not be optimal for diverse data sources. PF2LoRA proposes a two-level LoRA architecture: a common adapter shared across all clients and a personalized adapter for each client. PF2LoRA automatically determines a suitable rank for each individual client based on their local data, thereby tailoring adaptation to data heterogeneity. Experiments on natural language understanding and generation tasks are presented, aiming to demonstrate PF2LoRA's superior performance compared to existing federated fine-tuning methods.

 **Strengths**: Overall, reviewers acknowledged that the paper is generally well-written and easy to follow. Some reviewers also highlighted the idea of the two-level LoRA formulation and the automatic rank learning mechanism as a novel approach to address data heterogeneity.

**Weaknesses**: Reviewers raised **significant and numerous concerns**, primarily regarding the **technical novelty, experimental scope, theoretical foundation, and computational efficiency**:
*   **Limited Technical Novelty and Missing Related Work**: Multiple reviewers (R_b8Qd, R_4oqs, R_YmR9) pointed out that the core concept of using global and client-specific adapters in FL with LoRA is not new and has been explored in prior works like FedDecomp, and FedDPA.
*   **Methodological Clarity and Hyperparameter Sensitivity**: The explanation of how the algorithm achieves dynamic rank was deemed **not well articulated** (R_YmR9). The selection of new hyperparameters (e.g., client-adapter rank $r_k$) was described as **heuristic, lacking principled guidance** (R_yZVy). The ablation study on bilevel optimization needed more detailed analysis (R_yZVy). The specific choice of AdamW and SGD optimizers was questioned (R_YmR9).
*   **Insufficient Computational/Communication Efficiency Analysis**: Reviewer R_b8Qd noted a lack of comprehensive analysis. Reviewer R_yZVy highlighted that PF2LoRA incurs the **highest computational cost per communication round** (Table 18), posing a significant trade-off that could limit deployment on resource-constrained devices.
*   **Weak Theoretical Analysis**: Reviewers (R_yZVy, R_YmR9) noted that the theoretical analysis is **limited to a single-machine setting (M=1)**, making it largely unrelated to the proposed method in the federated learning context.
*   **Experimental Limitations and Baselines**: Experiments are **limited to smaller models** (RoBERTa, GPT-2), raising questions about scalability to larger LMs, among others.

**Decision**: The paper received consistently negative ratings (4, 4, 4, 2). Numerous and substantial weaknesses regarding novelty, experimental rigor, theoretical soundness, and practical considerations were raised. Crucially, the authors did not provide a rebuttal to any of the reviewer comments or questions. Thus, the paper, in its current form and without a rebuttal, does not meet the acceptance bar for ICLR.

**Reviewer Concerns:**

No rebuttal was provided; thus, all the concerns were not addressed.

**Reviewer Scores:**

Given the complete absence of a rebuttal from the authors, it is highly probable that all reviewers would have maintained their negative scores.

---

### Decision · Program_Chairs · 2026-01-26

Reject